# Reasoning on a Spectrum:
# Aligning LLMs to System 1 and System 2 Thinking

## Abstract

Large language models (LLMs) demonstrate remarkable reasoning capabilities, yet their reliance on step-by-step reasoning can make them brittle when tasks do not align with such structured approaches. In contrast, human cognition flexibly alternates between fast, intuitive reasoning (System 1) and slow, analytical reasoning (System 2), depending on context. To bridge this gap, we curate a dataset of 2K examples, each with valid responses from both reasoning styles, and explicitly align LLMs with System 1 and System 2 reasoning. Evaluations across diverse reasoning benchmarks reveal an accuracy-efficiency trade-off: System 2-aligned models excel in arithmetic and symbolic reasoning, while System 1-aligned models perform better in commonsense tasks. A mechanistic analysis of model responses shows that System 1 models employ more definitive answers, whereas System 2 models demonstrate greater uncertainty. Interpolating between these extremes produces a monotonic transition in reasoning accuracy, preserving coherence. This work challenges the assumption that step-by-step reasoning is always optimal and highlights the need for adapting reasoning strategies based on task demands.[1]

## 1 Introduction

LLMs have demonstrated remarkable reasoning capabilities, often achieving near-human or even superhuman performance (Huang and Chang, 2023). These advances have largely been driven by techniques that simulate step-by-step, deliberative reasoning, such as Chain-of-Thought (CoT) prompting and inference-time interventions (Wei et al., 2022b; Wang et al., 2022). Given their success, such methods are increasingly integrated into LLM training (Chung et al., 2024), reinforcing explicit, structured reasoning regardless of the task necessity. However, the increasing focus on step-by-step reasoning has revealed limitations such as brittle generalization, particularly in tasks requiring nuanced judgment (Delétang et al., 2023), logical consistency (Jiang et al., 2024), or adaptability to uncertainty (Mirzadeh et al., 2024). Similarly, recent analyses frame this issue as "overthinking": Cuadron et al. (2025); Chen et al. (2024) demonstrate that excessive deliberation can hamper decision-making. This problem appears in LLMs' responses to simple factual queries, where they often generate unnecessarily explanations instead of direct responses (Wang et al., 2023).

This focus on explicit, structured reasoning highlights a key difference between LLMs and human cognition: while LLMs are being pushed towards a single mode of processing, human reasoning is far more nuanced. Rather than a monolithic process, human reasoning emerges from a sophisticated suite of cognitive tools evolved to tackle a *spectrum* of computational problems. This spectrum of human reasoning encompasses both automatic and reflective processes, a key insight recognized across

---

[1]Our data and code are available at https://anonymous.4open.science/r/system12-CB8B

diverse fields from behavioral economics to psychology and neuroscience (Daw et al., 2005; Dolan and Dayan, 2013; Balleine and Dickinson, 1998). On one end lie computationally *light* problems demanding rapid, intuitive judgments (e.g., instinctively dodging a speeding car), handled by the reflexive "System 1." On the other end are *heavy* problems requiring deliberate, step-by-step analysis, managed by the reflective "System 2" (Kahneman, 2011; Stanovich and West, 2000). This dual-process system allows us to dynamically shift between modes depending on the task, balancing speed and accuracy (Evans and Stanovich, 2013). Extensive work in neuroscience in the past two decades links the dual-process framework and human decision strategies, which depicts decision-making on a spectrum between a fast but reflexive habitual decision strategy and a reflective goal-directed strategy (Daw et al., 2005; Dolan and Dayan, 2013). Experimental work in neuroscience is built on the relative advantages of these two strategies, the separate but overlapping neural structures supporting them, and the circumstances under which each system is deployed in the brain (Daw et al., 2011; Schad et al., 2020; Piray and Daw, 2021). Given the evolutionary advantage of humans in switching between fast and slow thinking to balance speed, efficiency, and accuracy, exploring LLMs through the lens of System 1 and System 2 reasoning offers a powerful way to address their current limitations.

While recent studies explore whether LLMs exhibit System 1 and System 2 behaviors (Hagendorff et al., 2023; Pan et al., 2024) or propose hybrid models (Yang et al., 2024; Deng et al., 2024), most prior work implicitly assumes that structured, deliberative reasoning is universally superior. Even research suggesting LLMs' capacity for both reasoning modes (Wang and Zhou, 2024) largely overlooks the crucial question of when each mode is indeed advantageous. The assumption that a single "best" reasoning strategy can apply across all contexts is a fundamental simplification that limits current approaches in LLM development. This assumption prevents LLMs from achieving true cognitive flexibility, hindering their ability to adapt their reasoning processes to diverse situations.

To address this gap, we explicitly align LLMs with System 1 and System 2 reasoning and evaluate their reasoning capabilities and behaviors across a range of reasoning benchmarks. Our approach involves designing an experimental setup where both thinking styles can produce valid responses but follow distinct paths, one leveraging intuitive heuristics, and the other prioritizing deliberate, step-by-step reasoning. By systematically assessing how reasoning styles and cognitive biases affect downstream task performance, we provide insights into when intuitive heuristics or structured deliberation are most effective, and highlight the trade-offs between accuracy and efficiency in LLMs.

Specifically, as demonstrated in Figure 1, we first curate a dataset of 2,000 reasoning questions, where each problem has both a fast, heuristic-driven (System 1) response and a deliberative, structured (System 2) response, grounded in 10 different cognitive heuristics (Tversky and Kahneman, 1974). We then explicitly align LLMs with either System 1 or System 2 type responses and evaluate these models on diverse reasoning benchmarks. Our findings reveal a structured accuracy-efficiency trade-off and demonstrate that different reasoning paradigms in LLMs excel at different types of tasks, mirroring how humans selectively rely on fast or slow thinking depending on task demands: System 2-aligned models consistently outperform instruction-tuned and CoT prompt baselines in arithmetic and symbolic reasoning, demonstrating superior multi-step inference, but generating more extended token-intensive responses. Conversely, System 1-aligned models generate more succinct responses and excel at commonsense reasoning, where heuristic shortcuts are effective. Importantly, unlike CoT models, which always engage in structured reasoning regardless of necessity, our models provide an explicit way to study when different reasoning styles are beneficial, mirroring the well-known efficiency-accuracy trade-off in human cognition (Keramati et al., 2011; Mattar and Daw, 2018). By framing LLM reasoning as a structured and adaptable process, rather than simply an ability to achieve higher benchmark scores, this work highlights the importance of selecting the right reasoning strategy for a given task. This perspective not only aligns LLM reasoning more closely with human cognition but also paves the way for more flexible, efficient, and robust reasoning systems, setting a foundation for future advancements in LLM reasoning.

## 2   Related Work

### 2.1   Reasoning in LLMs

Driven by extensive research highlighting the strengths and weaknesses of LLM reasoning abilities (e.g., Huang and Chang, 2022; Mondorf and Plank, 2024; Valmeekam et al., 2022; Parmar

et al., 2024; Sourati et al., 2024), recent efforts to enhance these capabilities have largely focused on prompting techniques (Brown et al., 2020), ranging from zero-shot prompting with explicit instructions (Kojima et al., 2022; Wang et al., 2023; Zhou et al., 2024b) to few-shot prompting with step-by-step examples (Wei et al., 2022b). Wang and Zhou (2024) take CoT prompting even one step further and demonstrate that CoT reasoning paths can be elicited from pre-trained LLMs by simply altering the decoding process without the use of a specific prompt. Related approaches, such as self-consistency decoding Wang et al. (2022), explore how diverse reasoning paths can enhance robustness, aligning with deliberative aspects of System 2 reasoning. Tree of Thought (ToT; Yao et al., 2024) generalizes over CoT and allows LMs to perform deliberate decision making by considering multiple different reasoning paths and self-evaluating choices to decide the next course of action, as well as looking ahead or backtracking when necessary to make a global choice. Another alternative way of increasing the reasoning abilities of LLMs is through instruction tuning on a substantial amount of CoT reasoning data Chung et al. (2024); Huang et al. (2022) or distillation Magister et al. (2022). By training LLMs on a large-scale CoT dataset, models can internalize step-by-step reasoning, potentially enhancing their performance across diverse benchmarks without relying solely on prompting techniques. Concurrent studies have identified an "overthinking" phenomenon in LLMs, where models produce excessively detailed or unnecessarily elaborate reasoning steps (Chen et al., 2024; Cuadron et al., 2025).

## 2.2 Dual-Process Theory in NLP

Dual-process theories, widely studied in psychology, distinguish between fast, intuitive reasoning (System 1) and slow, deliberate reasoning (System 2). While these theories have long explained the spectrum of human reasoning, their application in NLP remains underexplored. Existing research falls into two main categories: (1) analyzing LLMs' reasoning through dual-process theory, identifying similarities and differences between LLMs and human reasoning, and (2) developing models with dual-process mechanisms to enhance LLM reasoning and leverage the benefits of both systems.

**Analyzing LLMs' reasoning through dual-process theory.** Researchers have investigated whether LLMs exhibit reasoning behaviors aligned with System 1 and System 2, particularly in terms of cognitive human-like errors and biases (Hagendorff et al., 2023; Booch et al., 2021; Pan et al., 2024; Echterhoff et al., 2024; Zeng et al., 2024). Hagendorff et al. (2023) examine cognitive heuristics in LLMs, showing that newer models exhibit fewer errors characteristic of System 1 thinking. Booch et al. (2021) discuss fundamental questions regarding the role of dual-process theory in machine learning but leave practical implementation as an open problem. Most of these studies evaluate LLMs on benchmarks where System 2 reasoning is assumed to be superior, portraying intuitive responses as erroneous, even though such rapid, heuristic-driven judgments are often crucial for efficient and effective reasoning in real-world scenarios. In contrast, by analyzing models aligned with System 1 and System 2 reasoning using a carefully curated dataset where both response types are valid, we offer a more nuanced understanding of how this alignment influences broader model behavior.

**Incorporating dual-process theory in NLP models.** Several studies have integrated dual-process-inspired reasoning into LLMs. Some works combine intuitive (fast) and deliberate (slow) components to improve reasoning (He et al., 2024; Liu et al., 2022; Hua and Zhang, 2022; Pan et al., 2024), while others optimize reasoning efficiency by distilling System 2 insights into System 1 models (Yang et al., 2024; Deng et al., 2024; Yu et al., 2024). Additionally, research has leveraged System 2 reasoning to mitigate biases associated with System 1 heuristics, improving fairness and robustness (Furniturewala et al., 2024; Kamruzzaman and Kim, 2024; Weston and Sukhbaatar, 2023). While prior work largely frames System 2 reasoning as superior or explicitly builds dual-process components within models, our approach investigates the implicit effects of aligning LLMs to System 1 or System 2 responses. By analyzing how these heuristics influence general reasoning capabilities, we address a gap in the literature and provide new insights into the broader cognitive behaviors of LLMs that have implications for how unseen properties of data that LLMs are trained on can affect their capabilities.

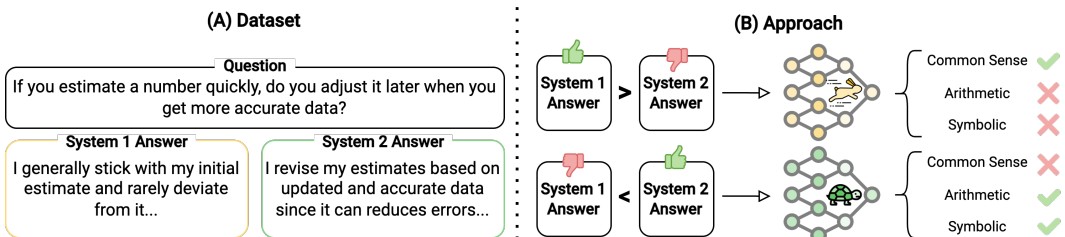

Figure 1: (A) Sample of dataset with System 1 and System 2 answers. (B) Overview of our alignment approach with fast and slow thinking, highlighting performance gains across reasoning benchmarks.

# 3 Method

## 3.1 Aligning LLMs to System 1 & System 2 Thinking

We formalize the modeling of fast and slow thinking as an alignment problem using a curated dataset in which each reasoning question is paired with both a System 1 (intuitive) and a System 2 (analytical) response (see Section 3.2). We align LLMs to either reasoning style via a preference-based training approach: for System 1 alignment, the intuitive response is designated as the preferred (winner) and the analytical response as the non-preferred (loser); for System 2 alignment, this preference is reversed, treating the analytical response as the winner and the intuitive response as the loser.

This approach is effective for two key reasons. First, our aim is not to introduce new knowledge or instructions but rather to shape the model's reasoning process based on existing capabilities. Second, previous research has shown that prompt engineering can guide LLMs toward System 2 reasoning (Wei et al., 2022a) or System 1 reasoning (Zhou et al., 2024a), suggesting that LLMs already have both reasoning abilities. Therefore, instead of creating new reasoning pathways, we guide the model to distinguish between intuitive and analytical reasoning processes without altering its underlying knowledge. The next section describes the dataset creation process that enables this training setup.

## 3.2 Dataset of System 1 & System 2 Thinking

Our curated dataset consists of 2,000 questions designed to elicit two distinct reasoning styles in English: one intuitive and rapid, reflecting cognitive shortcuts (System 1), and the other deliberate and analytical (System 2). This dual structure allows us to study the distinct mechanisms underlying System 1 and System 2 reasoning (Kahneman, 2011; Stanovich and West, 2000; Evans and Stanovich, 2013). The dataset was created in three key phases: Generation, Refinement, and Validation.

**Generation.** Cognitive heuristics provide a practical foundation for distinguishing between System 1 and System 2 reasoning, where both yield valid but behaviorally distinct responses (Kahneman, 2011). To construct our dataset, we adopted a human-in-the-loop pipeline that leverages GPT-4o (Hurst et al., 2024) to scale up the number of high-quality reasoning examples. In line with recent work on dataset creation using LLMs and few-shot prompting (Xu et al., 2023; Wang et al., 2022), we used a one-shot prompting setup, where each generation is guided by a carefully selected example grounded in a particular cognitive heuristic. These seed examples were authored by domain experts (see Appendix D) and span 10 well-known heuristics from Kahneman (2011) (Appendix C). For each, experts provided a reasoning question accompanied by both a System 1 (heuristic) and System 2 (deliberative) response. During expansion, the prompt included the formal definition of each heuristic, a description of how both systems typically approach it, and the expert-written example. This setup enabled the model to generate new reasoning items aligned with distinct cognitive patterns. Full prompt details are provided in Appendix F, and expert-authored examples are shown in Appendix E.

**Refinement.** As a byproduct of the data generation process, System 2 outputs were significantly longer and more detailed—reflecting their step-by-step reasoning style, while System 1 outputs were shorter and more direct; this length difference was confirmed using Welch's $t$-test, $t(2090.1) = -184.74$, $p < .001$, $d = -5.84$. Prior work demonstrates that alignment methods can rely on superficial cues, such as output length, favoring longer responses even when they offer no real reasoning advantage (Singhal et al., 2023). To prevent this bias, we use zero-shot prompting with

Table 1: Samples of curated dataset with the cognitive heuristic category, each paired with System 1 response using the heuristic and System 2 response mitigating that heuristic.

| Category | Question | System 1 Answer | System 2 Answer |
|---|---|---|---|
| Anchoring | When booking travel, do you find it hard to move away from the first deal you see? | I often find myself comparing everything to that initial deal. Once I've encountered a specific offer or price, it becomes the standard by which I measure all others. Even when new deals come along, my mind instinctively revisits that first one, guiding my judgment. It's an automatic comparison that influences how I evaluate options moving forward. | To book travel effectively, I avoid comparisons only against the initial option. I understand that the first deal isn't always the best ensures a broader perspective, more observations increases the chances of finding a more favorable travel plan. Exploring more options ultimately leads to better decision-making. |
| Halo Effect | Would you say fasting diet is effective? | This diet is obviously effective. just look at the people who follow it! Celebrities and influencers swear by it, and they're in great shape. When so many successful, healthy people use it, that's proof it works. Plus, it's super popular, which wouldn't happen if it didn't give great results. If you want to see real change, this is the way to go! | The effectiveness of a diet depends on scientific evidence, individual health factors, and long-term sustainability rather than its popularity or association with celebrities. A diet may lead to short-term weight loss but could lack essential nutrients or be difficult to maintain. Evaluating its impact on metabolism, overall health, and adherence is crucial.. |
| Over Confidence | Do you believe you will ace the interview? | Yes, I excel in interviews. I communicate clearly, stay confident under pressure, and listen attentively to questions. My ability to understand the interviewer's needs and align my responses accordingly enhances my effectiveness. I maintain engaging body language and make genuine connections, making a lasting impression. I prepare thoroughly, anticipate potential questions, and rehearse answers, ensuring I approach interviews with a calm, composed demeanor, making me a strong candidate. | While confidence in interpersonal skills is beneficial, thorough interview preparation is essential for success. It allows for anticipation of potential questions and crafting informed responses, showcasing an understanding of the company and role. Researching company culture enables candidates to align their answers with organizational values, enhancing their chances of making a positive impression. Solely relying on confidence can lead to unpreparedness, especially for technical inquiries, reducing the effectiveness of skill articulation. |

GPT-4o to match the lengths of our System 1 and System 2 outputs while preserving their content. Adjustments were applied only when there was a significant length disparity. More details about the prompt and the length disparity threshold are described in Appendix J. By reducing the length disparity, we minimized any preference for System 2 outputs arising from their longer responses. After adjustment, System 1 outputs had an average length of 82.19 tokens, while System 2 outputs averaged 83.93 tokens. A two one-sided t-test (TOST) confirmed the equivalence of post-adjustment lengths across various token counts as equivalence margins (see Appendix I), indicating that the adjustment effectively eliminated significant length differences between the two response types.

**Verification.** Prior works show that high-quality, expert-supervised datasets of this scale are common and effective for fine-tuning LLMs (Xiao et al., 2024; Dumpala et al., 2024; Li et al., 2024). Following this precedent to ensure data quality, we had our domain experts conform all generated data to formal definitions of System 1 and System 2 thinking, and ensured that the dataset covers the intended set of cognitive heuristics across varied subject areas. In this process, the experts manually revised approximately 20% of the responses. We further verified the breadth of topic coverage via topic modeling; see Appendix G for details. A subset of the curated dataset is shown in Table 1.

# 4 Experiments Setup

## 4.1 Alignment Algorithm

To implement the alignment strategy for System 1 and System 2 reasoning, we utilize two offline preference optimization methods, namely, Direct Preference Optimization (DPO; Rafailov et al., 2024) and Simple Preference Optimization (SimPO; Meng et al., 2024), because (i) their offline formulation removes the costly on-policy sampling loop, yielding a simpler and more compute-efficient training pipeline, and (ii) our hand-crafted preference pairs capture fine-grained relational signals that would likely be blurred by online-generated pairs.

DPO is an offline alignment method that fine-tunes LLMs by comparing the preferred and disfavored outputs of a model against a reference model, optimizing preferences without requiring a separate reward model. As a prominent method in preference optimization, DPO has gained traction for its stability and efficiency, making it a widely adopted alternative to Reinforcement Learning from Human Feedback (RLHF; Ouyang et al., 2022). SimPO builds on the principles of DPO but introduces a reference-free approach to preference optimization. Instead of requiring a separate reference model, SimPO aligns responses by directly optimizing preference signals within the model itself. This makes it computationally more efficient and removes the dependency on an external reference model, offering a streamlined alternative for aligning LLMs to a specific preference.

## 4.2 Benchmarks

We evaluate our System 1 and System 2 models using 13 reasoning benchmarks across three different categories: (1) arithmetic reasoning: MultiArith (Roy and Roth, 2015), GSM8K (Cobbe et al., 2021), AddSub Hosseini et al. (2014), AQUA-RAT (Ling et al., 2017), SingleEq (Koncel-Kedziorski et al., 2015), and SVAMP (Patel et al., 2021); (2) commonsense reasoning: CSQA (Talmor et al., 2019), StrategyQA (Geva et al., 2021), PIQA (Bisk et al., 2020), SIQA (Sap et al., 2019), and COM2SENSE (Singh et al., 2021); (3) symbolic reasoning: Last Letter Concatenation and Coin Flip Wei et al. (2022b). More details about the benchmarks are in Appendix H.

Following Kong et al. (2024), our evaluation follows a two-stage process. In the first stage, we present benchmark questions to model and record its responses. In the second stage, we prompt the model with the original question, its initial response, and benchmark-specific instructions to ensure the output is formatted as required. See Appendix K for each benchmark's instructions.

## 4.3 Implementation Details

We use Llama-3-8B-Instruct (AI@Meta, 2024) and Mistral-7B-Instruct-v0.1 (Jiang et al., 2023) as SFT models for alignment. Following Kojima et al. (2023), we compare the performance of these aligned models against their instruction-tuned counterparts under zero-shot and zero-shot CoT prompting (additional details in Appendix L). To analyze the model's behavior along the System 1 to System 2 reasoning spectrum, we train seven intermediate models, where the winner responses are mixed at predefined ratios between System 1 and System 2. This structured interpolation allows us to systematically assess whether the transition between reasoning styles is discrete or gradual.

# 5 Results

## 5.1 Distinct Strengths of System 1 & System 2 Models

Table 2 shows a comparison of exact matching accuracy across 13 benchmarks for Llama and Mistral. Specifically, we compare the base models with the System 1 and System 2 variants, and include results for CoT prompting for reference. Our findings reveal distinct performance trends for the System 1 and System 2 models, highlighting their respective strengths in different reasoning benchmarks.

In all arithmetic benchmarks (MultiArith, GSM8K, AddSub, AQuA, and SingleEq), System 2 models outperformed both the base model and their System 1 counterpart, evident for both Llama and Mistral. This improvement is most significant in the AddSub and SingleEq benchmarks. Similarly, System 2 models outperformed System 1 models in nearly all symbolic reasoning benchmarks (Coin, Letter), which require pattern recognition and logical structuring, further validating the idea that deliberative, slow-thinking models enhance performance in structured reasoning. While both approaches achieve high accuracy, System 1's heuristic shortcuts introduce small but systematic errors that System 2's deliberate, stepwise computations tend to avoid, such as rounding the number or adding numbers without checking. This is further supported by our AddSub analysis (see Appendix O).

Conversely, System 1 models excelled both their System 2 counterparts and the base model as well as the CoT variant on all commonsense reasoning benchmarks (CSQA, StrategyQA, PIQA, SIQA, COM2SENSE), which depend on intuitive judgments and heuristic shortcuts. While System 2 reasoning is correct, its deliberate nature can often lead to overthinking, producing overly cautious or extensively interpretive responses that diverge from typical human reactions in rapid, intuitive situations. For example, when asked what a kindergarten teacher does before nap time, System 2 suggests "encourage quiet behavior" instead of "tell a story," or predicts "laughter" rather than "fight" if you surprise an angry person. As shown in Appendix O, this preference for completeness over contextual fit makes System 2 less reliable for quick, socially grounded tasks.

When comparing Llama and Mistral, Llama models generally achieved higher accuracy across all benchmarks. This suggests that Llama may have stronger foundational reasoning capabilities, which are further enhanced by the System 2 and System 1 alignment. Moreover, instruction-tuned models equipped with the CoT prompt exhibited only marginal differences compared to their base counterparts because step-by-step reasoning has already been internalized during pretraining on CoT-style data (AI@Meta, 2024), reducing the need for explicit prompting. Based on this observation, we use the base Llama model as our primary baseline in subsequent experiments.

Table 2: Accuracy comparison of our System 1 and System 2-aligned models against instruction-tuned and CoT baselines across benchmarks. Each cell shows accuracy, with parentheses indicating the difference from the baseline. Color intensity reflects the magnitude of deviation.

| | | Arithmetic | | | | | | Symbolic | | Common Sense | | | | |
|---|---|---|---|---|---|---|---|---|---|---|---|---|---|---|
| | | MultiArith | GSM8K | AddSub | AQuA | SingleEq | SVAMP | Coin | Letter | CSQA | Strategy | PIQA | SIQA | COM2SENSE |
| System 2 | DPO | 98.67 (+1.0) | 79.37 (+0.88) | 89.87 (+7.4) | 49.21 (+0.39) | 94.37 (+3.65) | 85.4 (+4.9) | 93.8 (-0.4) | 86.2 (+2.2) | 71.42 (0) | 60.87 (-6.68) | 81.15 (-2.01) | 67.93 (-3.19) | 76.42 (-2.6) |
| System 2 | SIMPO | 97.83 (+0.16) | 79.38 (+0.89) | 90.13 (+7.66) | 54.72 (+6.78) | 94.49 (+3.77) | 81.7 (+1.2) | 94.4 (+0.2) | 84.8 (+0.8) | 69.62 (-1.8) | 67.38 (-0.17) | 81.49 (-1.67) | 69.16 (-1.96) | 78.21 (-0.81) |
| | Llama-3 | 97.67 | 78.49 | 82.47 | 48.82 | 90.72 | 80.5 | 94.2 | 84 | 71.42 | 67.55 | 83.16 | 71.12 | 79.02 |
| | Llama-3-CoT | 97.83 | 78.54 | 82.03 | 49.21 | 88.19 | 80.9 | 94.8 | 84.2 | 71.58 | 67.38 | 83.34 | 70.97 | 79.86 |
| System 1 | DPO | 98.5 (+0.83) | 77.01 (-1.48) | 80.76 (-1.71) | 46.46 (-2.36) | 77.24 (-13.48) | 78 (-2.5) | 93.4 (-0.8) | 83.8 (-0.2) | 72.81 (+1.39) | 68.21 (+0.66) | 83.94 (+0.78) | 72.16 (+1.04) | 79.99 (+0.97) |
| System 1 | SIMPO | 97.5 (-0.17) | 77.79 (-0.7) | 80.51 (-1.96) | 48.03 (-0.79) | 87.4 (-3.32) | 79.3 (-1.2) | 90 (-4.2) | 83.8 (-0.2) | 72.32 (+0.9) | 67.73 (+0.18) | 83.35 (+0.19) | 71.67 (+0.55) | 81.46 (+2.44) |
| System 2 | DPO | 78.83 (+1.16) | 56.45 (+1.47) | 81.27 (+6.79) | 32.68 (+1.19) | 84.84 (+0.98) | 69.1 (+3.4) | 41 (-2.2) | 8.6 (+8) | 62.82 (-3.44) | 56.81 (-8.6) | 80.49 (0) | 57.77 (-2.24) | 66.73 (-1.64) |
| System 2 | SIMPO | 78.3 (+0.63) | 55.42 (+0.53) | 82.28 (+7.8) | 34.25 (+2.76) | 86.81 (+2.95) | 68.5 (+2.8) | 45.4 (+2.2) | 7.8 (+6.2) | 64.78 (-1.48) | 63.75 (-1.66) | 82.07 (-0.46) | 59.82 (-0.19) | 68.15 (-0.22) |
| | Mistral | 77.67 | 54.89 | 79.75 | 31.49 | 83.86 | 66.26 | 43.2 | 1.6 | 66.26 | 65.41 | 82.53 | 60.01 | 68.37 |
| | Mistral-CoT | 78.3 | 54.96 | 80.25 | 33.07 | 83.66 | 67.8 | 43.8 | 1.6 | 66.18 | 65.49 | 82.21 | 60.76 | 69.01 |
| System 1 | DPO | 77.5 (-0.17) | 51.4 (-3.49) | 79.49 (-0.26) | 29.53 (-1.96) | 83.07 (-0.79) | 67.4 (-0.2) | 40.4 (-2.8) | 0 (-1.6) | 67.4 (+1.14) | 65.49 (+0.08) | 83.22 (+0.69) | 60.01 (0) | 70.83 (+2.46) |
| System 1 | SIMPO | 77 (-0.67) | 53.61 (-1.28) | 78.73 (-1.02) | 31.1 (-0.39) | 83.67 (-0.19) | 67.3 (-0.3) | 43 (-0.2) | 0 (-1.6) | 67.32 (+1.06) | 65.51 (+0.1) | 82.84 (+1.31) | 60.93 (+0.92) | 69.13 (+0.76) |

In summary, our results showcase that System 2 models excel in structured, multi-step reasoning such as arithmetic and symbolic reasoning, while System 1 models are effective in intuitive and commonsense reasoning benchmarks. These findings highlight the significant potential of dual-process alignment for boosting LLM performance across a diverse range of reasoning paradigms.

## 5.2 Length Differences Across Reasoning Styles

A recent trend in LLM performance, exemplified by models such as DeepSeek R1 (Muennighoff et al., 2025), is that achieving stronger benchmark results often correlates with producing longer reasoning chains, even if not explicitly trained to do so. This correlation raises the question of whether such verbose responses truly reflect enhanced reasoning capabilities or if they are simply a formatting artifact of current high-performing models. In our studies, this concern is particularly relevant for System 2 models, which are expected to behave more deliberatively. To investigate this, we analyze output lengths across the two-stage prompting setup described in Section 4.2.

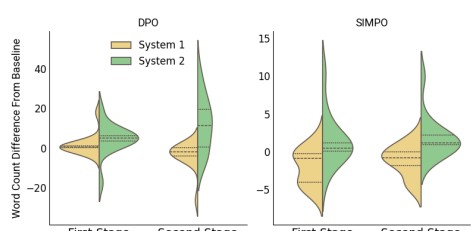

Figure 2: Token difference between System 1 and System 2 responses relative to Llama3 model across stages and alignment methods.

As shown in Figure 2, System 2-aligned models generate significantly longer responses than their System 1 counterparts, relative to the Llama baseline, under both alignment methods, DPO ($t(8836) = 57.14$, $p < .001$) and SimPO ($t(8586) = 9.833$, $p < .001$). This difference emerges specifically in the second stage, where models are prompted to finalize their responses, while response lengths remain comparable in the first stage, where both models are simply asked to reason. Although both models were trained on equal-length preference pairs (Section 3.2), System 2 models still tend to elaborate more during finalization, consistent with their alignment toward deliberative reasoning.

While longer reasoning chains are often associated with stronger performance, our findings suggest that this extended reasoning can also introduce inefficiencies or even degrade quality in contexts where concise, heuristic-driven reasoning is more appropriate. In particular, tasks requiring commonsense or intuitive judgments are often better handled by System 1 models, which respond more directly. This highlights a central insight of our study: extended reasoning is not universally beneficial, and reasoning strategies must be evaluated in relation to the task.

## 5.3 Moving from Fast to Slow Thinking

In the previous analysis, System 1 and System 2 models can be viewed as endpoints of a broader spectrum of reasoning strategies. Paralleling approaches in cognitive psychology (Daw et al., 2011;

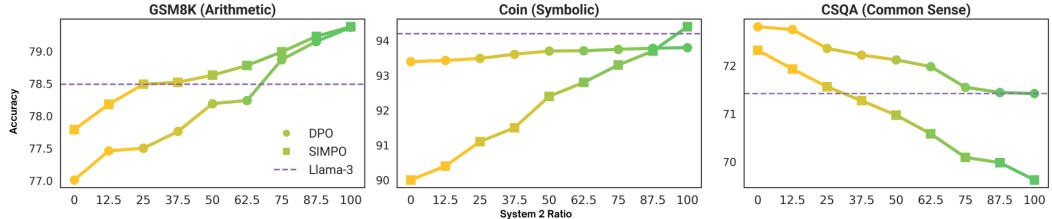

Figure 3: Accuracy across benchmark categories as reasoning shifts from System 1 to System 2.

Piray and Daw, 2021), we explored this spectrum by constructing interpolated models—blending System 1 and System 2 preferred answers at varying ratios in the alignment dataset. Figure 3 demonstrates a consistent, monotonic transition in accuracy across representative benchmarks from three reasoning categories (all $r^2 > 0.9, p < 0.001$), a pattern visible across all benchmarks (see Appendix M). While arithmetic and symbolic reasoning benchmarks exhibit a steady increase in accuracy moving toward System 2 thinking, commonsense reasoning benchmarks show the opposite trend, with accuracy increasing as models rely more on System 1 reasoning. This trade-off highlights that both reasoning styles offer unique advantages, with System 2 excelling in structured, multi-step problem-solving and System 1 providing efficient, adaptable responses in intuitive scenarios. These findings strengthen the importance of task-dependent reasoning strategies that leverage the strengths of both System 1 and System 2 thinking. Critically, there are no sudden drops or fluctuations in performance when transitioning between reasoning styles. This stability indicates that the shift from System 1 to System 2 reasoning is gradual and predictable, without any unexpected anomalies. This observation reinforces the idea that LLMs can be strategically guided toward different reasoning styles, allowing for more adaptive problem-solving.

## 5.4 Reasoning & Uncertainty

A key insight from psychology and neuroscience is that System 1 operates on confident heuristics, providing quick, intuitive judgments, while System 2 engages in more deliberate, analytical thought, accurately assessing the uncertainty associated with its conclusions (Daw et al., 2005; Lee et al., 2014; Keramati et al., 2011; Xu, 2021). To examine uncertainty and confidence, we consider three different characteristics: 1) token-level uncertainty; 2) the presence of hedge words in model output (Lakoff, 1973; Ott, 2018); and 3) definitive commitment to responses in System 1 versus System 2.

Plot A in Figure 4 shows that System 2 models consistently generate tokens with lower confidence than System 1 models, based on token-level uncertainty from logits. This trend holds across arithmetic $t(4075) = 54.53, p < .001$, symbolic $t(999) = 42.53, p < .001$, and commonsense $t(3510) = 106.86, p < .001$ benchmarks. Additionally, we analyzed surface-level uncertainty in model reasoning by examining word choices. Figure 4, Plot B shows System 2-aligned models use significantly more hedge words, in arithmetic $t(4075) = 22.03, p < .001$ and commonsense $t(3510) = 21.49, p < .001$ when models reiterate their reasoning. While increased uncertainty enhances analytical reasoning, it may hinder tasks requiring rapid, intuitive judgments. To assess early-stage response conclusiveness, we used LLM-as-Judge (Zheng et al., 2023) as detailed in Appendix N. Figure 4, Plot C shows System 1 models provide significantly more definitive responses than System 2 models in commonsense reasoning, *McNemar's* $\chi^2(1, 400) = 20.0, p < .001$, regardless of where in the response the definitive responses is reached (see Appendix N).

This analysis reinforces the idea that different reasoning styles are suited to different tasks. Greater uncertainty in models' generated reasoning suggests that System 2 models can explore alternative reasoning paths more effectively. This uncertainty is reflected in both their model output probabilities and word choices. System 2 models' superior performance in arithmetic benchmarks highlights the benefits of deliberate, effortful processing in tasks that demand exploration and uncertainty. On the other hand, the greater tendency of System 1 models to commit to responses in a more definitive way aligns with their advantage in tasks requiring rapid and intuitive judgments. This behavior is observed exclusively in commonsense reasoning, where quick, decisive responses are advantageous—a trend supported by human studies (Byrd, 2022) and confirmed by our findings in Section 5.1. However, it does not appear in other benchmarks (see Appendix N), suggesting that the activation of a particular reasoning style is context-dependent and influenced by task demands.

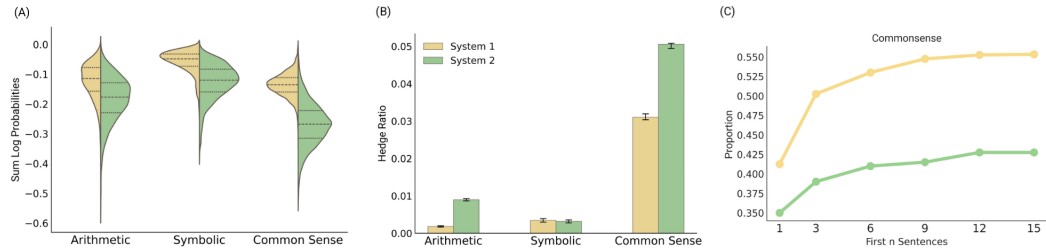

Figure 4: (A) Log probabilities of models' reasoning indicating internal uncertainty; (B) Hedge word ratio showing surface-level uncertainty; (C) Proportion of definitive answers in the first n sentences.

## 6 Conclusion

A central question in current LLM development is whether structured, step-by-step reasoning is always beneficial, or whether a more flexible range of reasoning strategies is needed. Inspired by dual-process theories of human cognition, we studied LLMs explicitly aligned with System 1 and System 2 thinking, representing fast, heuristic reasoning and slow, analytical reasoning, respectively. Our findings indicate that, much like in human cognition, reasoning in LLMs is not a one-size-fits-all solution: different reasoning modes are effective in different contexts and downstream tasks. System 2 excels in arithmetic and symbolic reasoning, while System 1 is more effective and accurate in commonsense reasoning (Section 5.1). Training intermediate models with blended ratios of preferred System 1 and System 2 responses revealed smooth, monotonic shifts in performance across benchmarks (Section 5.3), supporting the view that LLM reasoning lies on a continuous, tunable spectrum rather than a binary divide. Additionally, System 1 models generate responses with fewer tokens, highlighting its efficiency in decision-making (Section 5.2). Finally, our analysis in Section 5.4 illustrated that System 2 models exhibit greater uncertainty throughout the reasoning process, potentially enabling them to engage in more structured, step-by-step problem-solving. In contrast, System 1 models display higher confidence, allowing them to reach responses faster, which is particularly advantageous for tasks requiring rapid, intuitive judgments.

Beyond these empirical findings, our study aligns with broader principles observed across cognitive science and neuroscience. The observation that System 1 models generate faster responses echoes established theories in human cognition, where intuitive, heuristic-driven thinking allows for rapid decision-making. Similarly, the higher uncertainty exhibited by System 2 models aligns with neuroscience findings that deliberate reasoning involves increased cognitive load and self-monitoring mechanisms. These parallels suggest that LLMs, when properly aligned, can mirror key aspects of human cognition, offering new insights into both artificial and natural intelligence.

Our work bridges between LLM development and cognitive science, highlighting how we can enable efficiency-accuracy trade-offs in LLMs, similar to those long observed in human cognition. We align models with reasoning behaviors that follow well-known cognitive heuristics, which humans use in everyday thinking, like System 1's rapid, intuitive judgments and System 2's deliberate, analytical thought, and show they can follow the dynamic interplay between fast and slow thinking. This alignment not only informs more sophisticated training and evaluation strategies but also suggests that future LLMs can be designed to possess a more cognitively grounded flexibility, allowing them to adapt their reasoning as effectively as humans do when faced with diverse task demands. Finally, models that reason in ways that are cognitively interpretable, mirroring the human brain's strategies for learning, decision making, and inference, may also be more predictable, steerable, and trustworthy in deployment. In this light, dual-process alignment connects cognitive science and neuroscience with model capabilities, enabling future LLMs to reason more like humans, not just in what they conclude, but in how they get there.

This paper is a first step toward adaptive reasoning in LLMs, where models can dynamically shift between heuristic and deliberative thinking based on task demands. Furthermore, understanding how to optimally balance speed and accuracy in LLMs can have significant implications for real-world applications, from conversational agents to automated decision-making systems. In practice, this approach could let us deliberately trade off answer quality for faster responses by choosing fewer reasoning steps when time is critical.

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

## A    Limitations

Despite the promising advancements of using different thinking styles through the lens of dual-process cognitive theory in our approach, it is important to clarify the intended scope and outline future directions. Our curated dataset of 2,000 questions covers 10 well-established cognitive heuristics and was validated by our domain experts to ensure quality. While not exhaustive, this dataset provides a strong foundation for investigating reasoning style differences and establishes methodological groundwork for broader-scale expansion in future studies to represent the entire spectrum of reasoning challenges encountered in real-world tasks. We focused our alignment experiments on Llama and Mistral as base models, using DPO and SIMPO as preference optimization techniques. While our findings are likely to generalize across model architectures and alignment methods, given the shared emergence of both intuitive and deliberative reasoning in large-scale pretraining, testing this generalization to other architectures and alignment methods is a valuable future direction. In terms of evaluating reasoning uncertainty, we adopt token-level logit-based measures and linguistic hedging analysis as computationally tractable proxies. These provide interpretable signals of reasoning behavior, though deeper psycholinguistic and interactive evaluations may offer complementary insights. Finally, while our experiments reveal a clear accuracy-efficiency trade-off between intuitive and deliberative reasoning, the extent to which these findings translate to more complex or dynamic decision-making scenarios remains an open question. Future work should explore larger, more diverse datasets and investigate alternative alignment strategies to further validate and extend these results.

## B    Ethical Statement

Aligning LLMs with System 1 and System 2 reasoning raises concerns about model behavior in different contexts. System 1 models may produce overly confident but incorrect responses, while System 2 models, though more deliberate, may slow response times and increase computational costs. Responsible deployment requires balancing these trade-offs to prevent biased or misleading outputs.

## C    Cognitive heuristics

In Table 3, we list 10 different cognitive heuristics and their definitions, which we used in curating the dataset Kahneman (2011); Stanovich and West (2000); Evans and Stanovich (2013).

Table 3: 10 common cognitive biases and their definitions, which were considered in curating the dataset

| Cognitive Bias | Definition |
|---|---|
| Anchoring Bias | The tendency to rely too heavily on the first piece of information we receive about a topic, using it as a reference point for future judgments and decisions, even when new information becomes available. |
| Halo Effect Bias | The tendency to let one positive impressions of people, brands, and products in one area positively influence our feelings in another area. |
| Overconfidence Bias | The tendency to have excessive confidence in one's own abilities or knowledge. |
| Optimism Bias | The tendency to overestimate the likelihood of positive outcomes and underestimate negative ones. |
| Availability Heuristic Bias | The tendency to use information that comes to mind quickly and easily when making decisions about the future. |
| Status Quo Bias | The preference for maintaining the current state of affairs, leading to resistance to change. |
| Recency Bias | The tendency to better remember and recall information presented to us most recently, compared to information we encountered earlier |
| Confirmation Bias | The tendency to notice, focus on, and give greater credence to evidence that fits with our existing beliefs. |
| Planning Fallacy | The tendency to underestimate the amount of time it will take to complete a task, as well as the costs and risks associated with that task even if it contradicts our experiences. |
| Bandwagon Effect Bias | The tendency to adopt beliefs or behaviors because many others do. |

## D Details of Experts

The experts consulted are the two authors of this paper, both of whom are Ph.D. students in Psychology with a focus on cognitive and social science.

## E Initial Data Examples

Table 4: 10 samples generated by an expert

| Category | Question | System 1 Answer | System 2 Answer |
|---|---|---|---|
| Anchoring Bias | Do you rely on your first impression of meeting your lab mate ? | Yes, my gut instinct is usually right. | I should interact with them more to form a well-rounded opinion. |
| Halo effect Bias | How do you feel about the new political candidate? | I do not like their stance on one issue, so I think they are a terrible candidate. | I'll weigh their stance on multiple issues before deciding. |
| Over Confidence Bias | Do you think you will succeed in your new job? | I will definitely succeed here. | I will need to put in effort and adapt to the new environment to succeed. |
| Status Quo Bias | Should you change your workout routine? | My routine has always worked, so there is no need to change it. | My fitness needs might have changed, so I will consider adjusting my routine. |
| Optimism Bias | Do you need to double-check your work after a mistake? | I am usually careful, so one mistake doesn't mean I'll make another. | I will double-check my work to make sure I don't repeat the mistake. |
| Availability heuristic | Is the newest seafood restaurant the best restaurant in town? | It is the most popular one, so it must be the best. | Popularity does not always mean the best quality, so I will read reviews first. |
| Recency Bias | Should you invest in the stock after hearing good things about it? | Yes, it is been rising lately, so it's sure to keep going up. | I will research the stock and market conditions before making a decision. |
| Confirmation Bias | Is the newest seafood restaurant the best restaurant in town? | It is the most popular one, so it must be the best. | Popularity does not always mean the best quality, so I will read reviews first. |
| Planning Fallacy | Is the newest seafood restaurant the best restaurant in town? | It is the most popular one, so it must be the best. | Popularity does not always mean the best quality, so I will read reviews first. |
| Bandwagon Effect Bias | Why did you pick apple as brand of your phone? | Everyone I know has this brand, so it must be the best. | I compared different features and chose the one that suits my needs. |

The 10 samples generated by the expert for our data generation are shown in Table 4.

## F Prompt for Data Expansion

We expand our sample dataset by concatenating the expert-generated samples with the definitions in Table 3, along with a description of how System 1 and System 2 would respond to a given question, as shown below:

```
The System 1 response should be intuitive, fast, and reflect the cognitive
heuristic associated with the question.
```

```
The System 2 response should be more deliberate, slower, and use reasoning to
correct or mitigate the heuristic.
```

## G Topic Modeling

Following expert validation, we experimentally verified the diversity of our dataset to ensure it goes beyond surface-level variation in wording. Figure 5 presents the results of topic modeling using BERTopic (Grootendorst, 2022), demonstrating the range of topics covered in the dataset. The wide distribution and clustering across 150 unique topics demonstrate the semantic diversity of the dataset beyond superficial lexical variation.

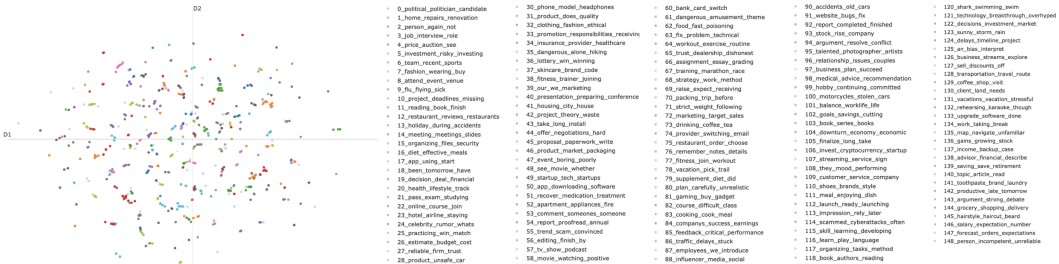

Figure 5: Topic modeling results on our dataset. Each dot represents a question, and colors indicate distinct topics.

## H   Benchmark Details

We use three categories of reasoning benchmarks: arithmetic, commonsense reasoning, symbolic reasoning, We provide an overview of the datasets used in each category.

**Arithmetic reasoning.**   We use six datasets: MultiArith, GSM8K, AddSub, AQuA, SingleEq, and SVAMP. Each dataset consists of questions that present a scenario requiring numerical computation and multi-step reasoning based on mathematical principles.

**Commonsense reasoning.**   To assess commonsense reasoning, we utilize five benchmarks: CommonsenseQA (CSQA), StrategyQA, PIQA, SocialIQA (SIQA), and Com2Sense. All require models to go beyond surface-level understanding and reason using prior knowledge. CSQA focuses on multiple-choice questions grounded in general world knowledge, while StrategyQA includes questions that demand implicit multi-hop reasoning. PIQA evaluates physical commonsense by requiring models to choose the more plausible solution to everyday benchmarks. SIQA targets social commonsense, presenting scenarios about interpersonal interactions and asking questions about motivations, reactions, and emotions. Com2Sense provides pairs of complementary sentences to test a model's ability to distinguish between plausible and implausible statements using commonsense.

**Symbolic reasoning.**   We use the Last Letter Concatenation and Coin Flip datasets. Last Letter Concatenation involves forming a word by extracting the last letter of given words in order. Coin Flip presents a sequence of coin-flipping instructions and asks for the final coin orientation. These datasets were originally proposed by Wei et al. (2023a) but were not publicly available. Kojima et al. (2023) later followed their approach to create and release accessible versions, which we use in our experiments.

## I   Equivalence Testing of Dataset Lengths Using TOST

A two one-sided t-test (TOST) confirmed the equivalence of these post-adjustment lengths across various token counts as equivalence margins: $\pm 3$ tokens, $t(3870.30) = 85.82$, $p < .001$; $\pm 5$ tokens, $t(3870.30) = 149.07$, $p < .001$; $\pm 7$ tokens, $t(3870.30) = 212.31$, $p < .001$; and 5% of the mean token count ($\pm 4.15$ tokens), $t(3870.30) = 122.29$, $p < .001$

## J   Length Adjustment Threshold and Prompt

We adjust the length if there is a disparity of more than 15 tokens between the System 1 and System 2 outputs using GPT-4o with the following prompt:

```
For a given {question}, we have two types of answers:
A fast, intuitive response based on cognitive heuristics which is our System
1 Answer.
System 1 Answer:  {System 1 Answer}
And a slow, deliberate, and logical reasoning response which is our System 2
Answer.
System 2 Answer:  {System 2 Answer}
Your task is to adjust the two answers so that they are presented in the same
order of tokens without altering their content.  Ensure that the intuitive
nature of the System 1 Answer and the logical reasoning of the System 2
Answer are preserved.
```

## K   Benchmark Instruction

The benchmark-specific instructions are shown in Table 5.

Table 5: Benchmark instruction sentences

| Benchmark | Second Stage Instruction |
|---|---|
| MultiArith, SingleEq, AddSub, GSM8K, SVAMP | Therefore, the answer (arabic numerals) is |
| AQuA, CSQA | Therefore, among A through E, the answer is |
| SIQA | Therefore, among A through C, the answer is |
| PIQA | Therefore, among A and B, the answer is |
| COM2SENSE | Therefore, the answer (TRUE or FALSE) is |
| Strategy, Coin | Therefore, the answer (Yes or No) is |
| Letters | Therefore, the final answer is |

## L   Implementation Details

We use Python 3.10.12, PEFT 0.12.0, PyTorch 2.4.0, and Transformers 4.44.2. The dataset is split
into 80% training and 20% validation. For alignment, we apply Low-Rank Adaptation (LoRA Hu
et al., 2021) with a rank of 8, an alpha of 16, and dropout rate of 0.1. We train for five epochs, using
accuracy on winner responses as an early stopping criterion to prevent overfitting, with patience of 5.
We set the train batch size to 4 and the validation batch size to 8. To align Llama 3 using the DPO
method, we followed Meng et al. (2024) and set the learning rate to $7e-7$ with beta of $0.01$. For
SimPO, we use a learning rate of $1e-6$, beta of $2.5$, and a gamma-to-beta ratio of $0.55$. For Mistral
v0.1, we set the DPO learning rate to $5e-7$ with beta of $0.001$. In SimPO, we use a learning rate of
$5e-7$, beta of $2.5$, and a gamma-to-beta ratio of $0.1$.

The experiments were conducted using NVIDIA RTX A6000 GPU equipped with 48GB of RAM.
The total computation time amounted to approximately 800 GPU hours.

## M   Moving from Fast to Slow Thinking Plots

Figure 6 demonstrates a consistent, monotonic increase in accuracy across all other benchmarks.

## N   Additional Insights into Models' Reasoning

In this analysis, we investigate when different models reach definitive answers. We aim to detect
this commitment as early as possible during the reasoning process. This early commitment serves
as a proxy for the model's confidence in the generated reasoning and its final answer. By analyzing
this behavior, we explore whether models can arrive at a definitive answer or if they leave room for
ambiguity or subjective interpretation.

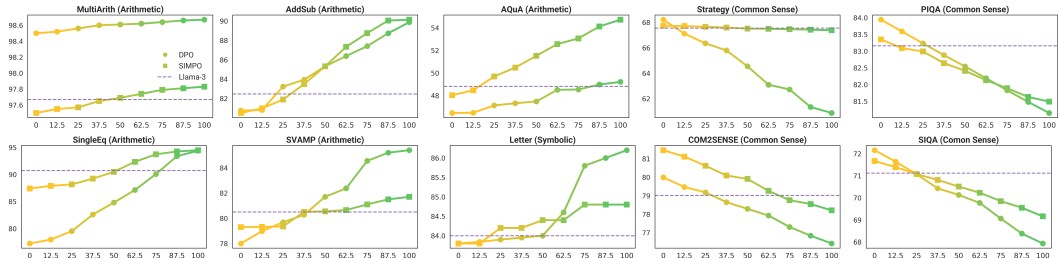

Figure 6: Accuracy across different benchmarks as reasoning shifts from System 1 to System 2.

We leverage the strong extractive capabilities of LLMs (Wei et al., 2023b) and their near-human-like annotation abilities (Gilardi et al., 2023; Alizadeh et al., 2023). Specifically, we focus on the Phi4 (14B) model (Abdin et al., 2024), which demonstrates exceptional performance in question-answering and reasoning benchmarks, even surpassing closed-source models like GPT-4o (Hurst et al., 2024). To determine whether a model's reasoning contains a definitive answer, we use the following prompt fed to Phi4:

> Does the given answer directly answer the given question in a definitive way? ONLY RETURN YES OR NO IN A \textbf{}. Definitive answers are clear and do not leave room for interpretation or ambiguity. If the answer tries to explore multiple perspectives or factors involved, it is not definitive, and YOU HAVE TO RETURN NO.

This prompt is applied to reasoning generated by both System 1 and System 2 models. To understand when these models commit to a definitive answer during their reasoning process, we focus on the first $n$ sentences of their reasoning, where $n \in \{1, 3, 6, 9, 12, 15\}$. We set a cap of 15 sentences based on our observations that nearly all generated reasonings across benchmarks fall within this range (see Figure 8).

Applying the prompt to each generated reasoning from the models across all benchmarks (200 randomly sampled data points from each benchmark, totaling 2000 samples for both System 1 and System 2 reasonings), we append six solved demonstrations to the prompt to help further guide the models. These demonstrations, selected randomly from the cognitive heuristics introduced in Section 3.2, help clarify what qualifies as a definitive answer, aligning the models' knowledge with patterns we have aligned System 1 and 2 models with (see Section 3.1).

Figure 7 shows the proportion of definitive answers in the first n sentences, across all benchmarks.[2] For tasks where quick, intuitive judgments are advantageous, such as in commonsense reasoning, System 1 models consistently provide more definitive answers than System 2 models. This gap emerges early, with System 1 providing more definitive answers in the first three sentences. The difference persists even as we extend the number of sentences considered (see Table 6 for a quantitative analysis of the significance between System 1 and System 2 regarding the definitiveness of their answers).

## O   System-Specific Failure Patterns

To complement the main results, we include two analyses that illustrate how System 1 and System 2 models diverge in failure patterns depending on task type. In numerical reasoning benchmarks, System 2 models are more reliable when higher precision is required, while in commonsense benchmarks, System 1 models tend to produce more contextually appropriate answers. The following figure and table offer additional insight into these differences.

To further analyze the behavioral differences between System 1 and System 2 models, we examine their performance on AddSub items with varying numeric complexity. Figure 9 shows the distribution of digit types in ground truth answers across four outcome categories. Notably, in examples where System 2 succeeds and System 1 fails ("Sys2 better"), the ground truth answers tend to have a

---

[2]Note that this ratio should not necessarily converge to 1.0 as more sentences are considered. In some cases, even when considering the full reasoning chain, the models may still leave room for vagueness.

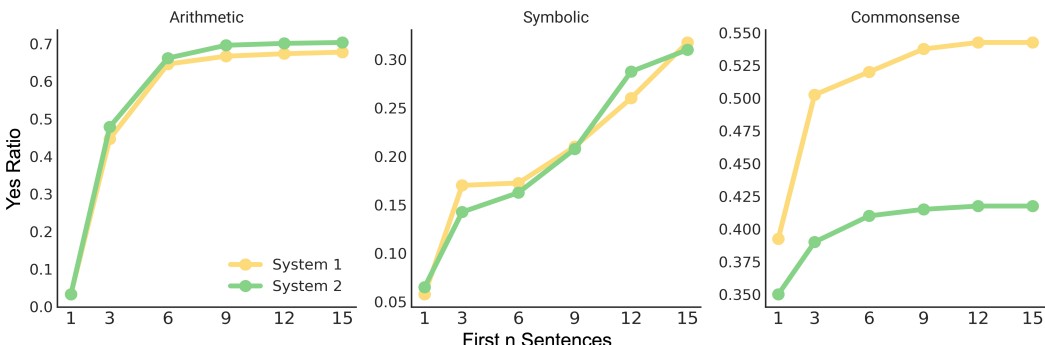

Figure 7: Proportion of definitive answers in the first n sentences across arithmetic, symbolic, and commonsense reasoning tasks

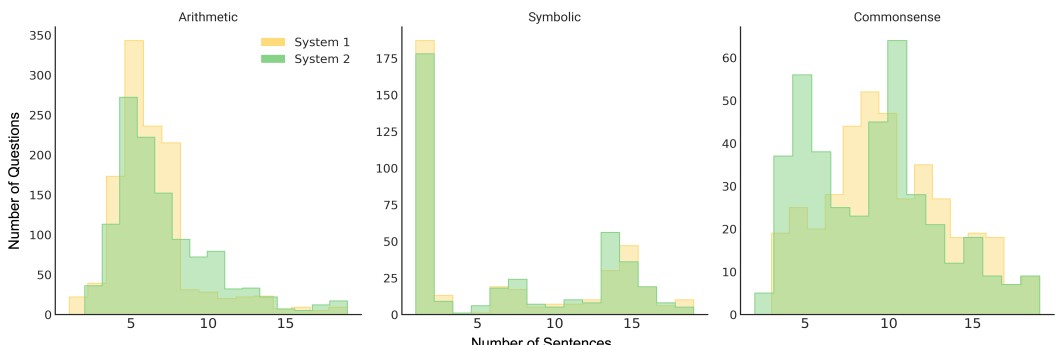

Figure 8: Distribution of the number of sentences in models' reasoning for both System 1 and System 2 reasoners across different benchmarks.

Table 6: McNemar's test results comparing the ratio of answers providing committed and definitive responses between System 1 and System 2 across different benchmarks. Statistically significant results ($p$-value < 0.05) are boldfaced.

| # Sen. | Arithmetic | | | Symbolic | | | Common Sense | | |
|---|---|---|---|---|---|---|---|---|---|
| | $\chi^2$ | $p$-value | Winner | $\chi^2$ | $p$-value | Winner | $\chi^2$ | $p$-value | Winner |
| 1 | 21.0 | 1.00 | System 1 | 19.0 | .755 | System 2 | 25.0 | **.050** | **System 1** |
| 3 | 123.0 | **.028** | **System 2** | 29.0 | .228 | System 1 | 20.0 | **> .001** | **System 1** |
| 6 | 125.0 | .272 | System 2 | 33.0 | .720 | System 1 | 21.0 | **> .001** | **System 1** |
| 9 | 120.0 | **.040** | **System 2** | 44.0 | 1.00 | System 1 | 21.0 | **> .001** | **System 1** |
| 12 | 118.0 | .051 | System 2 | 45.0 | .320 | System 2 | 20.0 | **> .001** | **System 1** |
| 15 | 121.0 | .069 | System 2 | 45.0 | .836 | System 1 | 20.0 | **> .001** | **System 1** |

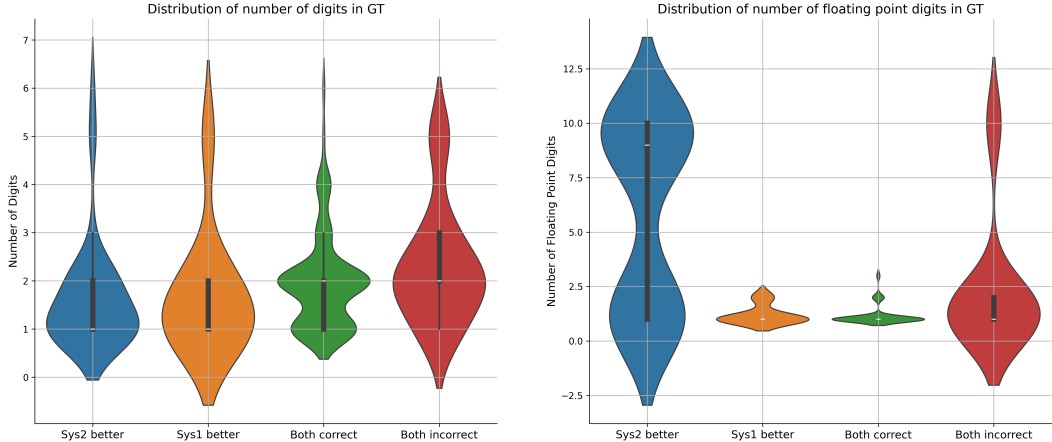

Figure 9: Distribution of digit characteristics in AddSub ground truth (GT) answers across different outcome categories. Left: total number of digits. Right: number of digits after the decimal point. Notably, examples where System 2 is correct and System 1 is not ("Sys2 better") tend to have more floating-point digits, indicating higher numerical precision.

significantly higher number of floating-point digits (Mann–Whitney U test, $U = 346.0$, $p = 0.0051$). This pattern suggests that System 2 is more effective at handling cases requiring greater numerical precision. In contrast, the number of total digits (irrespective of decimal placement) does not differ meaningfully between the "Sys2 better" and "Sys1 better" subsets ($U = 224.0$, $p = 0.99$).

We also provide a qualitative comparison of commonsense failures made by System 2, shown in Table 7. The table includes representative examples from CSQA where System 2 responses, although logically coherent, miss intuitive or socially grounded answers. These cases highlight how interpretive depth can lead to answers that diverge from typical human judgment.

# P    Technical Appendices and Supplementary Material

Technical appendices with additional results, figures, graphs and proofs may be submitted with the paper submission before the full submission deadline (see above), or as a separate PDF in the ZIP file below before the supplementary material deadline. There is no page limit for the technical appendices.

Table 7: Examples where the System 2 model fails on commonsense questions. The middle column shows the model's rationale and final answer, while the right column contains our analysis of why these responses deviate from typical human judgment. In all cases, the System 1 model gives the correct answer.

| Question (Answer Choices) | System 2 Rationale & Final Answer | Analysis and Insight |
|---|---|---|
| A supermarket is uncommon in what type of collection of shops? (A) strip mall (B) city or town (C) shoppingcentre (D) boutique (E) vermont | Supermarkets are most commonly found in urban or suburban areas such as cities and towns. They are less associated with specific retail arrangements like strip malls or boutiques. **Final Answer: B** | Overanalyzes spatial patterns instead of recognizing that "strip mall" is a retail form not typically associated with supermarkets. Correct answer: A. |
| The man tried to reply to the woman, but he had difficulty keeping track of conversations that he didn't do what to? (A) initiate (B) ignore (C) question (D) answer (E) ask | If someone has trouble following conversations, it's often because they are not actively participating by responding. **Final Answer: D** | Logical but misses the commonsense interpretation that initiating a conversation anchors attention. Correct answer: A. |
| What does a kindergarten teacher do before nap time? (A) lower expectations (B) encourage (C) fear (D) time test (E) tell story | Teacher might try to calm children and create a relaxed atmosphere before nap, e.g., by encouraging quiet behavior or mindfulness. **Final Answer: B** | Misses the stereotypical, culturally familiar act of storytelling before naps. Correct answer: E. |
| Where can a human find clothes that aren't pants? (A) pants shop (B) on planet earth (C) dress shop (D) school (E) train wreck | On planet Earth, people have access to a wide range of clothing items beyond pants, including dresses, skirts, and robes. **Final Answer: B** | Too literal and expansive—technically true but ignores the specific, expected answer: C. |
| Dan was sitting quietly with a book. Laurie thought he was focused, but he actually did what? (A) eat (B) think (C) reading (D) meditate (E) fall asleep | The word "ditting" seems like a typo for "reading," which matches the context of holding a book. **Final Answer: C** | Tries to correct perceived error in the question instead of interpreting the pragmatic cue. Correct answer: E. |
| What instrument can be played with an air of happiness? (A) jump up and down (B) jump up and down (C) sing (D) play games (E) fiddle | Singing is often associated with expressing happiness, and it's a way people naturally convey joy. **Final Answer: C** | Interprets "instrument" metaphorically rather than literally, overlooking that "fiddle" is an actual musical instrument. Correct answer: E. |
| If you spend all your time buying and not saving what is likely to happen? (A) using money (B) feel better (C) ocean (D) losing money (E) go broke | Without saving, continuous spending will likely cause one to lose money over time. **Final Answer: D** | Reasoning is valid, but the phrasing "losing money" is less idiomatic than "go broke," which better matches the question. Correct answer: E. |
| Surprising an angry person could lead to what? (A) humor (B) fight (C) jocose (D) laughter (E) accidents | A sudden surprise may defuse tension and even cause laughter, helping to de-escalate anger in some cases. **Final Answer: D** | Overly optimistic and detached from real-world expectations; more likely outcome is aggression. Correct answer: B. |

