# OpenReview forum: "Reasoning on a Spectrum: Aligning LLMs to System 1 and System 2 Thinking"
_NeurIPS.cc/2025/Conference — Submitted to NeurIPS 2025_

### Official Review · Reviewer_8XdA · 2025-06-09

**Clarity:** 3
**Significance:** 2
**Originality:** 2
**Rating:** 4
**Confidence:** 5

**Summary:**

This paper uses 2K LLM-synthesised data to align SFT models with System 1 (fast, intuitive) and System 2 (slow, analytical) thinking separately. With the resulting "System 1 models" and "System 2 models", the paper reports their performance differences across 13 reasoning benchmarks and argues the task specific performance divergence.
The paper also claims that System 2-aligned models generate longer responses than System 1-aligned models, even when trained on equal-length preference pairs. Interpolating between System 1 and System 2 alignment ratios reveals a monotonic transition in accuracy.
System 2 models exhibit higher uncertainty, while System 1 models provide more definitive responses early in their reasoning. Overall the paper challenging the "Step-by-Step Always Optimal" assumption.

**Questions:**

1. Is the effectiveness of the alignment training verifiable? Is there a justifiable basis to claim that training on system 1 thinking data of your system can transform an SFT model into a "system 1 model", and that training on system 2 thinking data can make it a "system 2 model"?
2. System 1 and system 2 thinking are typically distinct strategies for addressing different problems. Is it not arduous to curate both system 1 and system 2 thinking for the same question and compel the model to conform to them? I question the effectiveness and practicality of the alignment experiments. As we know there are new reasoning models that can conduct both thinking and direct answers (Qwen3, Claude 3.7) in the industry, how is your method compared to the industry practices? Are you following the path of these industry models?
3. You present numerous claims that are generally heuristically recognized by the community, yet they lack robust evidence. In the section "Length Differences Across Reasoning Styles", you conclude that “extended reasoning is not universally beneficial” without conducting experiments and statistical analyses to substantiate this claim. In fact, not every experiment in your paper necessarily supports your conclusion. As mentioned above, I have reservations about the "system 1/2 model", as well as the interpolated models. Additionally, I would like to see more elaborate details regarding the uncertainty experiments.
4. Recommended citation: Break the chain: Large language models can be shortcut reasoners

**Ethical Concerns:**

["NO or VERY MINOR ethics concerns only"]

**Limitations:**

Oversimplified theoretical framework
Modest data scale and synthetic nature
Gaps in experimental design and validation

**Quality:**

3

**Strengths And Weaknesses:**

Strenghts:
The paper is well organized and proficiently written, with no obvious errors.
The experiments are easily reproducible.
The cognitive science behind is well acknowledged.
The related work section is satisfying.

Weaknesses:
The data size is modest, the data is LLM generated, potentially limiting coverage of real-world reasoning complexity.
Aligns only with Llama and Mistral; results may not generalize to other well-studied and recognized LLMs (e.g., GPT-family models, Qwen series, Deepseek).
Assumes System 1/2 as discrete endpoints, but human reasoning often involves parallel processing rather than strict sequential switching. The paper’s linear interpolation may oversimplify cognitive dynamics.
It lacks details of the experiment setups, like how do the system 1/2 models comparison following the rules of contrast experiments

---

> ### Author Response · Authors · 2025-08-04
>
> We thank the Reviewer for their thoughtful critique and for acknowledging the clarity, organization, and cognitive science grounding of our work. Below, we address the key concerns raised.
>
> **Weakness Response**:
> 1. We appreciate the Reviewer’s concern regarding dataset scale and synthetic generation. Recent studies [1,2,3] have demonstrated that small, carefully curated, and expert-verified datasets can be highly effective for aligning LLM behavior. Although the initial examples were LLM-generated, every example in our dataset was reviewed and verified by domain experts. Around 20% of the examples were revised to ensure that the intended cognitive distinctions between System 1 and System 2 reasoning were clearly and faithfully represented. While a dataset of 2,000 examples may appear modest by current LLM standards, our primary focus was on alignment quality rather than data volume. As discussed in lines 625–629, we explicitly acknowledge this limitation
> 2. We agree that including a broader range of models can further strengthen generalization However, evaluating alignment across all foundation models was out of the scope of this study.  We deliberately focused on open-source, instruction-tuned models (LLaMA-3 and Mistral). Our method is model-agnostic and can be applied to any other model. As noted in the Limitations section (lines 629–633), we acknowledge this constraint and clarify that generalization to other model families is an important direction for future work.
> 3. The relationship between System 1 and System 2 is indeed more nuanced than discrete endpoints. Neuroscience reveals an active debate about how reflective versus deliberative processes are implemented in the brain, with some theories emphasizing separate processing (arguing that these systems have mutually exclusive benefits), while others focus on concurrent processes. Influential studies in humans [4,5] as well as in rats [6] have demonstrated behavior best explained by a hybrid combination of reflective and reflexive systems, and our approach is consistent with these studies. Our work does not assume a binary view but models System 1 and System 2 as interpretable endpoints on a reasoning spectrum. Moreover, LLMs are inherently autoregressive and operate sequentially. Therefore, modeling fully parallel reasoning in such architectures demands fundamental architectural changes and is beyond the scope of this paper. We focus instead on establishing a tractable and controlled framework for aligning LLMs to distinct reasoning styles, leaving more complex integration of concurrent reasoning for future work.
> 4. We acknowledge that our linear interpolation strategy is a simplification of the richer dynamics involved in human cognition. Our goal was not to fully capture the complexity of cognitive switching, but to provide an interpretable and systematic way to explore how reasoning behavior shifts between System 1 and System 2 alignment. We will clarify this point further in the camera-ready version. As shown in Section 5.3, even this simple interpolation reveals smooth, monotonic transitions in task performance and uncertainty, suggesting that LLMs can be guided along a reasoning spectrum.
> 5. We respectfully disagree with the reviewer on this comment. Our methodology follows a classic experimental design where two distinct "treatments" are compared against a shared "control" group to isolate the effects of our alignment strategies. In our setup, the base instruction-tuned models (Llama-3-Instruct and Mistral-7B-Instruct-v0.1) serve as the control group. The two treatment groups are the models explicitly aligned to prefer either System 1 or System 2  responses from our curated dataset. To ensure a valid and controlled comparison, we held all other key variables constant across the groups. Specifically, both System 1 and System 2 models within a family (e.g., Llama) begin from the same base model, are trained using identical preference optimization algorithms (DPO and SimPO), use the same training dataset size, and are evaluated with the exact same benchmarks and two-stage prompting procedure. The sole independent variable that differentiates the two treatment groups is the alignment signal—that is, whether the System 1 or the System 2 response was designated as the preferred "winner" during training. Furthermore, to mitigate potential confounding factors, we systematically controlled for response length in our preference dataset to ensure alignment targeted the reasoning style itself rather than superficial cues. This rigorous design allows us to directly attribute the observed differences in performance, efficiency, and uncertainty to the specific cognitive alignment strategy applied, providing a clear contrast between the two reasoning styles and the baseline.

---

> > ### Author Response · Authors · 2025-08-04
> >
> > **Question Response**:
> > 1. Yes, in our work the effectiveness of alignment is supported both by quantitative performance trends and qualitative behavioral differences. As described in Section 5.1, System 1-aligned models consistently outperform System 2 models on intuitive, commonsense benchmarks, while System 2-aligned models excel on arithmetic and symbolic tasks requiring multi-step reasoning. Beyond accuracy, we conduct extensive qualitative analyses in Sections 5.2, 5.4, and Appendices N and O. System 1 models tend to produce shorter, more confident responses, often making early, decisive judgments, whereas System 2 models generate longer, more uncertain outputs and display deliberative traits such as hedge words and explicit step-by-step reasoning. These patterns are consistent across models (LLaMA-3 and Mistral) and alignment methods (DPO and SimPO), demonstrating that our alignment process reliably induces distinct and verifiable reasoning behaviors.
> > 2. We respectfully disagree with the reviewer. Both systems are, actually, routinely deployed for the same tasks in human cognition. Experimental work suggests that time pressure, stress, and cognitive load lead the reflexive system to be deployed, while calm conditions and promising reward promote the reflective system [7,8,9,10]. Consider the question “Is this person trustworthy?” System 1 immediately produces a gut feeling based on facial features and body language ('something feels off'), while System 2 analyzes evidence like credentials, past behavior, and logical consistency. Both responses are natural and valid approaches to the same question. Recent LLM development, however, has overwhelmingly focused on promoting slow, deliberative reasoning (e.g., CoT), often leading to "overthinking" where models produce excessively complex responses for simple queries. Our decision to create a dataset with both System 1 and System 2 answers for each question was a deliberate design choice to counteract this imbalance. This dual-response structure, developed through a rigorous three-stage process of generation, refinement, and expert validation, creates the controlled setting necessary to systematically study the trade-offs between these two cognitive modes, which would not be possible if they were only applied to different problems.
> > Regarding large industry models like Qwen3 or Claude 3.7, while they may produce both thoughtful and direct answers, their proprietary nature makes it impossible to determine if this behavior stems from an explicit cognitive alignment strategy like ours. In contrast, our framework provides transparent, explicit control over the cognitive alignment process. This allows for a reproducible scientific analysis of how different thinking styles are instantiated, revealing a tunable spectrum of reasoning with predictable trade-offs between accuracy and efficiency. Therefore, rather than following industry practices, our work provides a foundational methodology to analyze and steer the cognitive behaviors that may only be implicitly present in proprietary models.
> > 3. We respectfully challenge the characterization of our claim—that "extended reasoning is not universally beneficial"—as a widely accepted community heuristic. In fact, this is an active and emerging area of investigation, with several recent studies beginning to identify an "overthinking" phenomenon where excessive deliberation can be detrimental to performance [11,12]. Our work contributes directly to this new line of inquiry by providing a set of controlled experiments and statistical analyses specifically designed to substantiate this claim. Our results demonstrate this task-dependent trade-off clearly: while our System 2 models produce longer responses, they are significantly less accurate on all five commonsense reasoning benchmarks compared to the more succinct System 1 models. Furthermore, our interpolation experiments provide robust statistical evidence for this relationship, showing a smooth, monotonic decrease in commonsense reasoning accuracy as models are increasingly aligned with more deliberative System 2 reasoning. These findings provide direct, quantitative support for our conclusion. Regarding the request for more details on the uncertainty experiments, we provide a comprehensive analysis in Section 5.4 and Appendix N. We use three distinct metrics: 1) token-level logit uncertainty, 2) the frequency of hedge words, and 3) a LLM-as-Judge evaluation for response definitiveness. As detailed in the paper, each of these analyses includes rigorous statistical testing (e.g., Welch’s t-tests, McNemar’s χ² tests), revealing significant and consistent differences between the reasoning styles across various benchmarks.

---

> > > ### Author Response · Authors · 2025-08-04
> > >
> > > [1] Xiao et al. "HealMe: Harnessing Cognitive Reframing in Large Language Models for Psychotherapy." Proceedings of the 62nd Annual Meeting of the Association for Computational Linguistics (Volume 1: Long Papers). 2024.
> > >
> > > [2] Dumpala et al. "Sugarcrepe++ dataset: Vision-language model sensitivity to semantic and lexical alterations." Advances in Neural Information Processing Systems 37 (2024): 17972-18018.
> > >
> > > [3] Li et al. "Language grounded multi-agent reinforcement learning with human-interpretable communication." Advances in Neural Information Processing Systems 37 (2024): 87908-87933.
> > >
> > > [4] Daw et al. "Model-based influences on humans' choices and striatal prediction errors." Neuron 69.6 (2011): 1204-1215.
> > >
> > > [5] Gillan et al. "Characterizing a psychiatric symptom dimension related to deficits in goal-directed control." elife 5 (2016): e11305.
> > >
> > > [6] Miller et al. "Dorsal hippocampus contributes to model-based planning." Nature neuroscience 20.9 (2017): 1269-1276.
> > >
> > > [7] Daw et al. "Uncertainty-based competition between prefrontal and dorsolateral striatal systems for behavioral control." Nature neuroscience 8.12 (2005): 1704-1711.
> > >
> > > [8] Keramati et al. "Speed/accuracy trade-off between the habitual and the goal-directed processes." PLoS computational biology 7.5 (2011): e1002055.
> > >
> > > [9] Shenhav et al. "Toward a rational and mechanistic account of mental effort." Annual review of neuroscience 40 (2017): 99-124.
> > >
> > > [10] Kool et al. "Mental labour." Nature human behaviour 2.12 (2018): 899-908.
> > >
> > > [11] Chen et al. "Do not think that much for 2+ 3=? on the overthinking of o1-like llms." arXiv preprint arXiv:2412.21187 (2024).
> > >
> > > [12] Cuadron et al. "The danger of overthinking: Examining the reasoning-action dilemma in agentic tasks." arXiv preprint arXiv:2502.08235 (2025).

---

### Official Review · Reviewer_TS6F · 2025-06-30

**Clarity:** 3
**Significance:** 2
**Originality:** 2
**Rating:** 4
**Confidence:** 3

**Summary:**

This submission presents a framework for examining and influencing the reasoning of Large Language Models (LLMs) utilizing dual-process theory from cognitive science. The authors compile a dataset comprising 2,000 questions, each accompanied by two legitimate responses that reflect rapid, intuitive "System 1" cognition and deliberate, analytical "System 2" reasoning. Utilizing this dataset, they adjust LLMs (Llama-3 and Mistral-7B) to each reasoning style via preference optimization techniques (DPO and SimPO). The submission further analyzes the characteristics of these models, demonstrating that System 2 reasoning is associated with longer, more uncertain responses, while System 1 is more token-efficient and definitive.

**Questions:**

1. Have you done any early tests to make a "router" model that picks between the System 1 and System 2 experts based on a question?  Or maybe a single, larger model could be made to choose its own mode of operation?
2. Is it possible to look at the models' internal states to support your claim that their "thinking" styles are similar?  For example, do attention patterns or activation vectors show consistent differences between the System 1 and System 2 models when they are processing the same input?
3. Does the conclusion from this submission, such as "System 2 models exhibit greater uncertainty throughout the reasoning process, potentially enabling them to engage in more structured, step-by-step problem-solving," also apply to the reasoning model, such as QwQ-32B?

**Ethical Concerns:**

["NO or VERY MINOR ethics concerns only"]

**Final Justification:**

This submission provides detailed discussion and experimental results to balance the system-1 and system-2 thinking strategy, which is important for the efficient deployment of LLM, especially for the large reasoning model. In summary, this submission is worth being presented at the conference. Thus, I recommend accepting it.

**Limitations:**

yes

**Quality:**

3

**Strengths And Weaknesses:**

**Strengths:**

- The submission looks at accuracy in more than just surface-level ways.  By using interpolation to look at the reasoning spectrum, the trade-offs in token efficiency, and the study of model uncertainty (through logits, hedge words, and response definitiveness), we can learn a lot about how the alignment process changes behavior.
- The submission tests 13 different benchmarks using a variety of models, such as Llama-3 and Mistral-7B, as well as alignment methods like DPO and SimPO.  These benchmarks include tasks that require arithmetic, common sense, and symbolic reasoning, which makes the differences between System 1 and System 2 reasoning clear and reliable.
- The paper is well-organized.  And it's easy to see the difference between LLM step-by-step reasoning and System 1/2 thinking.

**Weaknesses:**

- The submission’s core methods primarily integrate existing preference optimization techniques (e.g., DPO, SimPO) and dual-process cognitive theories without proposing algorithmic frameworks. The strategies for organizing and aligning the dataset are systematic; however, the overall technical approach depends on the integration of existing methods, suggesting a deficiency in innovative methodological advancements that could improve its algorithmic contribution.
- While the submission adeptly demonstrates the existence of a reasoning trade-off, it does not provide a mechanism or discussion for a model to dynamically select the appropriate reasoning style for a particular task.  This constrains the immediate practical effect of the work.
- The validity of certain "System 1" examples in the dataset may be scrutinized. In some cases, the answers seem to be more like commentaries on how to use intuition (for example, "My gut instinct is usually right" in Table 10, Appendix E) than direct, intuitive outputs. This could make the model articulate System 1 thinking instead of executing it, which is a subtle but important difference.
- The analysis of uncertainty serves as a preliminary step in this investigation, yet the primary emphasis is on aligning the textual output.  The issue of whether the model's foundational computational mechanism fundamentally alters remains predominantly unexamined.  More in-depth investigations (e.g., examining internal activations) are required to validate the assertion of aligning cognitive processes.

---

> ### Author Response · Authors · 2025-08-04
>
> We thank the Reviewer for the thoughtful and constructive feedback. We appreciate the recognition of our analysis beyond accuracy, the clarity of our System 1/2 framing, and the breadth of our benchmark evaluation.
>
> **Weakness Response**:
> 1. We appreciate the Reviewer’s point that our approach builds on existing preference techniques and cognitive theories. We agree that our primary contribution is not a new optimization algorithm, but rather a novel framework that bridges cognitive science and LLM alignment.  Specifically, this framework integrates existing techniques to enable new capabilities and analyses, successfully aligning LLMs with dual-process theory. To our knowledge, this is the first work that has systematically compared the benefits and shortcomings of different thinking styles in LLMs using a method grounded in human cognitive heuristics. Furthermore, our work offers novel methodological components, such as the construction of a high-quality dual-response dataset and the demonstration of a tunable reasoning spectrum via interpolation. While we used established alignment methods, these components and findings offer meaningful new tools for analyzing and steering LLM reasoning, which we believe opens the door for further exploration in this direction.
> 2. We completely agree with the Reviewer’s insightful observation regarding the potential to dynamically select between reasoning styles, as noted in Lines 637–639, we explicitly acknowledge this as a promissing direction for future research. While this paper focuses on establishing a foundation by aligning LLMs independently to System 1 and System 2 reasoning styles and analyzing their distinct strengths and trade-offs, we agree that dynamic reasoning represents a valuable next step but out of the scope of this paper. Specifically, we are exploring an extension of this work that incorporates an agentic framework with a lightweight controller or meta-model capable of selecting between reasoning modes based on task characteristics.
> 3. We believe there is a confusion here: Table 10 in Appendix E contains only the initial seed examples, which were manually constructed by experts to bootstrap the dataset creation process. These examples are not part of the final dataset used for model alignment. As described in Section 3.2, the full dataset was constructed through a three-stage pipeline, generation, refinement, and expert validation, resulting in 2,000 crafted examples that reflect distinct reasoning strategies. A representative sample of the final data is shown in Table 1.
> 4. Our paper's central contribution is a novel framework for aligning with System 1 and System 2 reasoning, with our analysis focusing on the consequential *behavioral effects* of this alignment, such as changes in accuracy, efficiency, uncertainty, and response definitiveness. We agree with the Reviewer that probing internal model activations is a valuable and complementary direction for future research, a point we acknowledge in lines 635–636. While a mechanistic analysis would undoubtedly offer additional insights, we believe that establishing and validating these distinct reasoning behaviors is a robust and necessary first step. A deeper investigation into the model's internal computational mechanisms, while an important next step, was therefore beyond the scope of this paper.

---

> > ### Author Response · Authors · 2025-08-04
> >
> > **Question Response**:
> > 1. We appreciate this suggestion and fully agree that dynamic selection of reasoning style is a compelling direction for future work, as noted in our response to Weakness 2 and in the lines 637–639. Our current study focuses on establishing a foundation by aligning models independently to System 1 and System 2 reasoning styles and analyzing their distinct strengths and trade-offs. This was an intentional first step to enable systematic evaluation of their strengths, trade-offs, and behavioral signatures. We are exploring an extension of this work that incorporates an agentic framework with a lightweight controller or meta-model capable of selecting between reasoning modes based on task characteristics.
> > 2. We agree this is a crucial point. Our approach is analogous to cognitive psychology, where the focus is on characterizing rich, observable behaviors to understand cognition. As discussed in lines 635–636 and in our response to the Weakness 4, we analyze the distinct behavioral signatures of our aligned models, such as the trade-offs in accuracy and efficiency, or differences in uncertainty. Just as behavioral findings in psychology ground neuroscience, establishing these robust patterns provides the necessary foundation and hypotheses for future mechanistic work that probes internal activations. Therefore, while analyzing internal states is a key direction for future work we acknowledge in our paper, our contribution here is to first provide this essential behavioral map.
> > 3. Yes; concurrent papers [1,2,3] have explored how reasoning language models express and manage uncertainty, particularly in the context of step-by-step reasoning. They investigate how uncertainty manifests during reasoning and how it can be used to improve calibration and model behavior. This aligns with our findings that System 2 models exhibit greater uncertainty throughout the reasoning process, potentially enabling them to engage in more structured, step-by-step problem-solving.
> >
> > [1] Mei et al. "Reasoning about Uncertainty: Do Reasoning Models Know When They Don't Know?." arXiv preprint arXiv:2506.18183 (2025).
> >
> > [2] Yoo, et al. "Reasoning models better express their confidence." arXiv preprint arXiv:2505.14489 (2025).
> >
> > [3] Yin et al. "Reasoning in flux: Enhancing large language models reasoning through uncertainty-aware adaptive guidance." Proceedings of the 62nd Annual Meeting of the Association for Computational Linguistics (Volume 1: Long Papers). 2024.

---

> > > ### Comment · Reviewer_TS6F · 2025-08-05
> > >
> > > Thank you for providing a detailed discussion. All my concerns are well-addressed. I will raise my score to 4.

---

> > > > ### Author Response · Authors · 2025-08-05
> > > >
> > > > Thank you for reconsidering your score. If there is anything we can further clarify or improve that might support a higher score, we would be grateful for your suggestions.

---

### Official Review · Reviewer_mUhz · 2025-07-01

**Clarity:** 4
**Significance:** 3
**Originality:** 3
**Rating:** 5
**Confidence:** 4

**Summary:**

This paper proposes a novel framework for aligning large language models (LLMs) with human-like reasoning styles inspired by dual-process theories from cognitive psychology—namely, System 1 (fast, intuitive, heuristic) and System 2 (slow, analytical, deliberate) thinking. The authors construct a carefully curated dataset of 2,000 questions, each paired with both System 1 and System 2 style answers, grounded in ten canonical cognitive heuristics. Using preference-based fine-tuning (DPO and SimPO), they train LLMs to preferentially adopt one reasoning style over the other. The paper empirically demonstrates that System 2-aligned models perform better on structured, multi-step tasks (e.g., arithmetic), while System 1-aligned models excel in commonsense reasoning tasks, offering faster and more intuitive responses. The authors also analyze reasoning behaviors (length, certainty, hedge usage), revealing smooth interpolations along the reasoning spectrum and highlighting a trade-off between accuracy and efficiency.

**Questions:**

The most important question I have concerns a potential **conceptual and ethical tension** raised by the findings:

> Your results show that aligning LLMs to System 1 reasoning—characterized by stronger cognitive biases—leads to better performance on commonsense reasoning benchmarks. This suggests that injecting heuristic, bias-prone reasoning may improve accuracy in some tasks.
>

This leads to two key concerns:

1. **Interpretability of the Result**: Could the authors provide more clarification on *why* stronger cognitive biases appear to help on commonsense tasks? Do these benchmarks inherently reward human-like biases? If so, to what extent is this a desirable signal versus an artifact of benchmark design?
2. **Risk of Misuse or Misinterpretation**: There is a risk that future work may interpret this as justification to intentionally align models toward biased behavior to improve perceived "commonsense" performance. How do the authors recommend mitigating this risk, especially when such biases may be harmful in real-world applications (e.g., stereotype amplification, unfair judgments)?

**Ethical Concerns:**

["NO or VERY MINOR ethics concerns only"]

**Limitations:**

**Yes.**

The authors provide a comprehensive and thoughtful discussion of their work’s limitations. They explicitly address the behavioral side effects of aligning models to either System 1 or System 2 reasoning styles—including verbosity, uncertainty, and potential heuristic bias—and analyze these effects quantitatively (Section 5.4). They also acknowledge the limitations of their dataset scale and the specificity of their benchmark-driven evaluation, noting that results may not fully generalize to interactive or real-world scenarios. Furthermore, the paper’s conclusion and ethical impact section (Section 6, Broader Impacts) reflect a clear awareness of the societal risks associated with over-relying on a single reasoning style, particularly System 1, in contexts requiring precision and fairness.

Given these reflections, I believe the authors have been appropriately transparent, and there are no major omissions in their self-assessment.

**Paper Formatting Concerns:**

Page 22 (NeurIPS Checklist - Limitations): There is a broken LaTeX reference (“We discuss the limitation of the paper in the ??”) that should point to a specific section, such as Appendix A. This is a minor formatting error but should be corrected before final submission.

**Quality:**

4

**Strengths And Weaknesses:**

Strengths:

- **Quality**: The experimental design is rigorous, well-controlled, and grounded in both machine learning methodology and cognitive science theory. The authors take great care to control for confounding factors such as response length and structure.
- **Clarity**: The paper is well-written and clearly structured. Key concepts—such as dual-process theory, preference optimization, and reasoning trade-offs—are introduced with sufficient background for both NLP and cognitive science audiences.
- **Originality**: This work represents a unique interdisciplinary contribution, bridging cognitive psychology and LLM training in a way that is rarely explored. Explicitly aligning LLMs with different cognitive styles—rather than assuming step-by-step reasoning is always superior—is a fresh and valuable angle.
- **Significance**: The findings challenge prevailing assumptions in LLM research (e.g., “more reasoning steps is always better”) and could influence future directions in model alignment, especially for applications involving user interaction, commonsense, or time-sensitive tasks.

Weaknesses:

- **Interpretability risks**: While System 1 alignment improves commonsense benchmarks, it also reinforces cognitive biases. This introduces a tension between performance and cognitive reliability that is not fully resolved, and risks being misinterpreted or misapplied by future practitioners. (See the Questions section for a detailed discussion)

---

> ### Author Response · Authors · 2025-08-04
>
> We sincerely thank the Reviewer for the detailed and positive assessment of our work, particularly for highlighting the rigor of our experimental design, clarity of presentation, and interdisciplinary contribution. We address the key concerns below.
>
> **Question Response**:
> 1. As prior cognitive science work [1,2,3,4,5] suggests commonsense reasoning tasks are more likely to engage System 1 processes. Our findings show that System 1-aligned models, which internalize stronger cognitive heuristics, excel on commonsense tasks. We want to clarify that the benchmarks were not selected based on whether they inherently reward human-like biases. Rather, based on prior work [6] we rely on well-established commonsense reasoning benchmarks in NLP to evaluate reasoning ability and these are not tailored to favor one reasoning style over another. Therefore, we do not view the differential performance of System 1 and 2-aligned models on different benchmarks as an artifact, but as an insight into the broader cognitive behaviors of LLMs.
> 2. We thank the Reviewer for raising this important concern and we agree that introducing heuristics could potentially lead to harmful biases. It should be noted, however, that there are significant evolutionary advantages to these heuristics, as they allow humans to reduce overthinking and make efficient judgments. Our use of heuristics in the paper is to gain this efficiency, not to justify or amplify harmful biases. We will clarify this distinction in the Ethical Statement of the camera-ready version of our paper. This addition will emphasize that while System 1-style heuristics can be appropriate in certain practical applications, they must be carefully managed to mitigate the risk of inadvertently creating biased or unfair outcomes. Therefore, a primary mitigation strategy involves creating adaptive models that can dynamically shift toward more deliberative, System 2 processing for high-stakes applications where fairness is critical, thereby leveraging the full, tunable reasoning spectrum we identified.
>
> We thank the Reviewer for pointing out the LaTeX reference issue on Page 22 in the NeurIPS Checklist. We will fix this in the camera-ready version.
>
>
>
>
> [1] Kahneman, D. (2011). Thinking, fast and slow. macmillan.
>
> [2] Evans et al. (2013). Dual-process theories of higher cognition: Advancing the debate. Perspectives on psychological science, 8(3), 223-241.
>
> [3] Gigerenzer et al. (1996). Reasoning the fast and frugal way: models of bounded rationality. Psychological review, 103(4), 650.
>
> [4] Gigerenzer et al. (2008). Fast and frugal heuristics are plausible models of cognition: Reply to Dougherty, Franco-Watkins, and Thomas (2008).
>
> [5] Dolan et al. "Goals and habits in the brain." Neuron 80.2 (2013): 312-325.
>
> [6] Kong  et al. "Better zero-shot reasoning with role-play prompting." arXiv preprint arXiv:2308.07702 (2023).

---

> > ### Comment · Reviewer_mUhz · 2025-08-05
> >
> > Thank you for the detailed explanation. All of my questions have been fully addressed.

---

### Official Review · Reviewer_STYE · 2025-07-03

**Clarity:** 2
**Significance:** 2
**Originality:** 2
**Rating:** 3
**Confidence:** 3

**Summary:**

This paper explores aligning LLMs with two types of human reasoning: fast, intuitive (System 1) and slow, analytical (System 2). Using a dataset of 2,000 examples reflecting both styles, the study shows that System 2 models perform better on arithmetic tasks, while System 1 models excel in commonsense reasoning. The results highlight the trade-off between accuracy and efficiency and suggest that reasoning strategies should adapt to task requirements.

**Questions:**

- Why is commonsense reasoning assumed to require rapid and intuitive judgments?
- Do the selected benchmarks faithfully reflect those characteristics?
- Are the models separately trained on System 1 and System 2 reasoning types and then applied to the corresponding benchmarks?

**Ethical Concerns:**

["NO or VERY MINOR ethics concerns only"]

**Limitations:**

yes

**Quality:**

2

**Strengths And Weaknesses:**

Strengths:
* This paper presents an interesting perspective by linking problem types to different reasoning styles (System 1 and System 2).
* The experimental results effectively support the proposed concept.
* Extensive analyses are provided, offering intuitive insights into model behavior.

Weaknesses:
* The connection between the benchmark categories and the corresponding reasoning styles (System 1 vs. System 2), as claimed by the author, appears insufficiently substantiated and requires further clarification. This should include a clear justification for why each type of benchmark aligns with a particular type of reasoning.
* The results seem somewhat unsurprising, as models trained to prefer a particular reasoning style are naturally more aligned with tasks that reflect that same style.
* In the generated dataset, both the questions and the System 1 and System 2 responses seem more like generalized descriptions of expected behavior for each reasoning type, rather than grounded, context-specific interactions. For instance, the question and the System 1 answer in the first row of Table 1 read more like an explanation of how that system typically behaves, rather than a realistic question paired with a natural, situated answer.
* A more thorough qualitative evaluation (e.g., output examples) is needed to clearly illustrate how the System 1 and System 2 models behave across different benchmark types. This would help verify whether the models’ performance improves in the intended manner.

---

> ### Author Response · Authors · 2025-08-04
>
> We thank the Reviewer for the thoughtful feedback and for highlighting the strengths of our work, particularly the alignment of reasoning styles to task types, the experimental evidence supporting this framework, and our detailed analyses of model behavior. Below, we address each of the concerns and questions raised.
>
>
> **Response to weakness**
> 1. We appreciate the Reviewer's concern regarding the substantiation of how benchmark categories align with System 1 and System 2 reasoning. We agree that there is no strict one-to-one mapping between each dataset and System 1 or System 2, and we have made this mapping based on prior literature [1,2,3,4,5], where commonsense reasoning tasks are more likely to engage System 1, whereas arithmetic and symbolic reasoning tasks primarily engage System 2. Specifically, commonsense tasks exhibit the hallmarks of System 1 thinking: they elicit automatic and rapid responses with minimal variability across individuals. We will revise the paper for the camera ready version to clarify this nuance. Our results show System 2-aligned models consistently outperform both baselines and System 1 models on arithmetic and symbolic tasks, while System 1 models outperform their counterparts on all commonsense reasoning benchmarks. To further support our findings we present a series of analyses, including output length comparisons (Section 5.2), performance interpolation across reasoning styles (Section 5.3), and reasoning uncertainty and definitiveness studies (Section 5.4, Appendix N).
> 2. To our knowledge, prior work has not systematically trained LLMs to reflect dual-process reasoning modes or evaluated their performance across diverse reasoning benchmarks. While it might seem intuitive that models aligned with a given reasoning style perform better on tasks reflecting that style, our work goes significantly beyond validating this intuition. As noted in lines 134–135, our goal is to analyze how aligning LLMs with cognitive heuristics affects broader reasoning behavior across tasks.
> 3. We acknowledge the Reviewer's concern that some responses in Table 1 resemble stylized reasoning paths. This was a deliberate design choice: our goal was not to mimic naturalistic human answers, but to provide clear, process-oriented examples of System 1 and System 2 reasoning. In early experiments, we found that outcome-focused examples, those providing only final answers in natural language, did not meaningfully influence model behavior. To address this, we focused on helping models internalize distinct reasoning strategies rather than surface-level answer patterns.  Therefore, this approach enabled models to internalize and exhibit the expected cognitive styles. This is supported by multiple analyses in Sections 5.2 and 5.4, and in Appendix O. We will clarify this motivation more explicitly in the camera-ready version to ensure our design rationale is clear.
> 4. Following the Reviewer's suggestion, we will include side-by-side System 1 and System 2 responses for each benchmark in the Appendix to further strengthen the paper, in the camera ready version. Below, we provide a representative sample from each category as an illustration.
>
> **Response to questions**:
> 1. As mentioned in our response to weakness 1, based on prior research [1,2,3,4,5] suggests that commonsense reasoning tasks are more likely to engage System 1 processes. We want to clarify that rather than making strong assumptions about the exact cognitive processes underlying each benchmark, we highlight our findings that System 1-aligned models consistently achieve higher accuracy on commonsense reasoning benchmarks and System 2-aligned models excel on arithmetic and symbolic reasoning.
> 2. We did not select benchmarks in order to faithfully reflect those characteristics, instead we selected well-established reasoning benchmarks which are widely used in NLP to evaluate reasoning. After aligning LLMs with distinct reasoning styles, we evaluated their performance on these benchmarks to analyze how aligning LLMs with cognitive heuristics influences broader LLM reasoning behavior. To further validate this mapping and characteristics, we conducted both quantitative and qualitative analyses across Sections 5.2, 5.3, and 5.4 and Appendices N and O.
> 3. Yes. We use an alignment setup (Section 3.1, Section 4.1) where models are separately fine-tuned to prefer either System 1 or System 2 responses. Once aligned, they are evaluated across all benchmark types.

---

> > ### Author Response · Authors · 2025-08-04
> >
> > | Benchmark category | Question | System 1 answer | System 2 answer |
> > |--------------------|----------|-----------------|-----------------|
> > | Arithmetic | Jason went to the mall on Saturday to buy clothes. He spent \\$14.28 on shorts and \\$4.74 on a jacket. In total, how much money did Jason spend on clothing? | Jason spent around \\$14 on shorts and around \\$4 on a jacket. That adds up to \textbf{18}. | Let’s break this down. Jason spent \\$14.28 on shorts and \\$4.74 on a jacket. To find the total amount he spent on clothing, we add the two amounts precisely. 14.28 + 4.74 = ? Start by adding the cents: 0.28 + 0.74 = 1.02. Then add the dollars: 14 + 4 = 18. Now combine: 18 + 1.02 = 19.02. The final answer is \textbf{19.02}. |
> > | Symbolic | Take the last letters of each word in "Lizzy Juany Aisha Brenda" and concatenate them. | Two names end in y, we group it as just one y and two other names end in a so one a. Therefore it is \textbf{ya}. | Let’s go step by step. First we identify the last letter of each word:Lizzy → y, Juany → y, Aisha → a, Brenda → a. Now concatenate all of them in order, including duplicates: y + y + a + a = yyaa. So, the final answer is \textbf{yyaa}. |
> > | Commonsense | Surprising an angry person could lead to what? (A) humor (B) fight (C) jocose (D) laughter (E) accidents | Surprising an angry person usually feels like it would make things worse, so the quick, gut answer is \textbf{(B) fight} | Although anger often leads to defensive or aggressive responses when surprised, it’s important to consider that human emotions are complex and context-dependent. A well-timed or unexpected surprise might catch the angry person off guard in a way that diffuses their tension instead of escalating it. In some cases, the surprise can trigger a moment of unexpected amusement or laughter, breaking the cycle of anger and fostering a brief moment of levity. Laughter has been shown to be a powerful social tool that can reduce hostility and promote reconciliation. Therefore, while the immediate reaction might seem like it would be a fight, it is also plausible, especially with the right kind of surprise, that the person could respond with laughter, leading to a positive outcome. Hence, \textbf{(D) laughter} is a valid and insightful answer reflecting the potential for surprise to transform anger into humor and reduce conflict. |
> >
> > [1] Kahneman, D. (2011). Thinking, fast and slow. macmillan.
> >
> > [2] Evans et al. (2013). Dual-process theories of higher cognition: Advancing the debate. Perspectives on psychological science, 8(3), 223-241.
> >
> > [3] Gigerenzer et al. (1996). Reasoning the fast and frugal way: models of bounded rationality. Psychological review, 103(4), 650.
> >
> > [4] Gigerenzer et al. (2008). Fast and frugal heuristics are plausible models of cognition: Reply to Dougherty, Franco-Watkins, and Thomas (2008).
> >
> > [5] Dolan et al. "Goals and habits in the brain." Neuron 80.2 (2013): 312-325.

---

> > > ### Comment · Reviewer_STYE · 2025-08-05
> > >
> > > Thank you for the response. While I appreciate the authors' additional efforts, the replies do not adequately address my concern regarding the connection between the thinking types, and the corresponding datasets / benchmarks (particularly given that some commonsense QA benchmarks require reasoning). Therefore, I will maintain my original score for the paper.

---

> ### Author Response · Authors · 2025-08-05
>
> Thank you for your engagement and highlighting the need for a clearer explanation. For further clarification, we would like to elaborate on how the System 1 and System 2 thinking maps onto the corresponding benchmarks.
>
> We emphasize that both System 1 and System 2 are well‑established forms of human reasoning [1,2]. They are not “reasoning” versus “non‑reasoning,” but two complementary cognitive modes. System 1 refers to fast, automatic, heuristic‑driven thinking which is efficient for many everyday and social judgments. System  2 refers to slow, deliberate, step‑by‑step reasoning that is more effortful and resource‑intensive, but often superior for structured, symbolic, or multi‑step problem solving.
> System  2 excels when careful sequential reasoning and error‑checking are required.
>
> Our benchmark categories reflect these theoretical distinctions. Commonsense reasoning tasks rely on contextual fit and heuristic shortcuts, aligning with System 1 strengths. Arithmetic and symbolic reasoning tasks require explicit intermediate steps, aligning with System 2 strengths.
>
> **An example from commonsense reasoning**
>
> Q: Surprising an angry person could lead to what? (A) humor (B) fight (C) jocose (D) laughter (E) accidents
>
> System 1: Quick, gut response: (B) fight.
>
> System 2: Considers alternative social dynamics: (D) laughter as a possible de-escalation.
>
> **An example from arithmetic**
>
> Q: Jason went to the mall on Saturday to buy clothes. He spent \\$14.28 on shorts and \$4.74 on a jacket. In total, how much money did Jason spend on clothing?
>
> System 1: Rounds and estimates: ~18.
>
> System 2:  computes exactly: =19.02.
>
> This mapping follows decades of cognitive science findings [1,2]. Our contribution is to test whether aligning LLMs with these modes yields analogous trade‑offs in performance.
>
> Importantly, our results show that models aligned to either System 1 or System 2 both achieve strong performance across all benchmarks. That is, both are effective, and neither alignment universally underperforms. Only in certain scenarios does one style show a mild superiority, which is precisely the nuanced trade-off we aim to highlight.
> The key message of our paper is not that one style is inherently better. Rather, we show empirically that a one‑size‑fits‑all reasoning strategy is suboptimal: aligning an LLM toward System 1 or System 2 shifts its strengths and weaknesses in predictable ways. This mirrors human cognition, where switching between modes based on context is critical to effective reasoning.
>
> We will clarify this mapping more explicitly in the camera-ready version.
>
> [1] Kahneman, D. (2011). Thinking, fast and slow. macmillan.
>
> [2] Evans et al. (2013). Dual-process theories of higher cognition: Advancing the debate. Perspectives on psychological science, 8(3), 223-241.

---

> ### Comment · Reviewer_STYE · 2025-08-06
>
> Let's not broaden the definition of reasoning in this context. I meant deliberate reasoning by reasoning. For example, I can easily find examples in StrategyQA (see Tables 1 and 2 in the paper) that require more than just System 1 thinking (i.e., fast, intuitive, heuristic responses).
>
> A more thorough review of commonsense benchmarks seems necessary.
>
> Following this discussion, I find myself leaning toward rejection (lowering my original score). Please let me know if I’ve misunderstood any key points.

---

> ### Author Response · Authors · 2025-08-06
>
> We sincerely apologize for any misunderstanding and agree with your point, tasks like StrategyQA clearly involve deliberate reasoning, and we do not claim otherwise.
>
> Our intent is not to suggest a rigid mapping between benchmark categories and reasoning types, but rather to explore whether different tasks tend to benefit more from the reasoning behaviors we encouraged in our System 1‑aligned and System 2‑aligned models. We chose the System 1/System 2 framing because these two broad reasoning styles (one leaning toward more intuitive reasoning and the other toward more step‑by‑step analysis) are well‑recognized in human cognition and reflect natural variation in how people approach problems. This framing offers a concise, recognizable way to study the impact of such variation in LLMs.
>
> That said, we want to be clear: we are not importing all properties of System 1/System 2 from cognitive science (e.g., “automatic” vs. “effortful” processing), nor are we implying that one is reasoning and the other is not. Our use of these terms is intentionally narrow, purely as an operational distinction for comparing two reasoning styles, not as a theoretical claim about the full dual‑process framework.
>
> In our experiments, the System 1‑aligned model tended to perform better on commonsense‑style benchmarks (including StrategyQA), while the System 2‑aligned model performed better on arithmetic and multi‑step logic tasks. We present this mapping as an empirical observation highlighting the role of different reasoning mechanisms across benchmarks, not as a categorical classification. This framing is similar to distinctions already in the LLM literature, such as OpenAI o3’s “reasoning effort” settings or recent work contrasting “less” vs. “more” overthinking.
>
> Our findings show that aligning LLMs toward different reasoning styles can lead to predictable, task‑specific performance shifts, even when both styles are genuine forms of reasoning and both achieve high performance across all benchmarks. This reinforces that our work is not about declaring one style “correct” and the other “incorrect,” but about demonstrating the value of reasoning diversity.
>
> Since our main contribution is not to claim that commonsense reasoning maps one‑to‑one with System 1 reasoning (or arithmetic with System 2), we would greatly value your opinion on the best way to frame this finding so it communicates our intent most clearly. How would you recommend we adjust our framing to prevent similar confusion for readers? We would be more than happy to incorporate any suggested changes to ensure our framing is fully aligned with our intended contribution.

---

### Author Response · Authors · 2025-08-04
**Apology and Clarification Regarding Author Response Submission**

Dear Reviewers,

Apologies for the delay in responding to your thoughtful reviews. We mistakenly submitted our replies under the “Author AC Confidential Comment” section instead of the official author response area. We only recently realized the error and wanted to extend our sincere thanks for your feedback, as well as clarify that our full responses were in fact prepared and submitted—just unfortunately in the wrong section.

We understand this may have caused confusion or appeared as a lack of engagement on our part, which was not our intention at all. We truly appreciate the time and effort you put into your reviews, and we regret the oversight.

Thank you again for your comments and consideration.

---

### Comment · Area_Chair_WvmK · 2025-08-05
**Rebuttal Issues**

All,

In this paper's case the responses were posted incorrectly and not available by the deadline -- the following policy is in place -- We regret that any rebuttals that were missing by the rebuttal deadline and were posted in their entirety after Jul 30 (e.g., by using the comment button) are to be ignored. Unfortunately, it is unfair on the vast amount of authors who “did the right thing” to count such late rebuttals.

If you can please respond that would be nice but we are being officially directed to not consider these rebuttals.

---

> ### Author Response · Authors · 2025-08-06
>
> Dear Area Chair,
>
> Thank you for clarifying the policy regarding rebuttal submissions. We completely understand the importance of fairness and sincerely apologize for our mistake in how the rebuttal was posted.
>
> To clarify, our responses were prepared and submitted to the area chair as an “Author AC Confidential Comment” before the July 30 deadline, as shown by the timestamps. The issue was with where we submitted, not with the timing or content, we did not gain any extra time or advantage in our mistake.
>
> We also received positive feedback from the reviewers, and one reviewer even increased their score following our response. While we fully acknowledge that the error was ours, we kindly ask if our case might be reconsidered given that our rebuttal was prepared and communicated on time.
>
> Thank you very much for your understanding and all your efforts.
>
> Best regards,

---

> > ### Comment · Area_Chair_WvmK · 2025-08-06
> >
> > You can appeal to the PC chairs if you want to but as I read the policy these comments were not posted correctly. Therefore as AC I have to follow the policy unless directed otherwise at least when it comes to getting reviewers to engage with the comments.

---

> > > ### Comment · Program_Chairs · 2025-08-07
> > >
> > > The AC is right. Rebuttal that was not posted through the official rebuttal interface shall be discarded. Thanks.

---

### Note · Authors · 2025-08-12

We want to clarify the circumstances regarding our rebuttal. Our responses to reviewers were submitted before the deadline (timestamps available). However, due to an unintentional error, we posted them as an "Author AC  confidential comment" visible to ACs by the deadline. PCs and ACs decided to discard our rebuttal. We sincerely apologize for this error and stress that no unfair advantage was gained. We appreciate reviewers' efforts and note that one reviewer increased their score, indicating that our clarifications were helpful in addressing concerns. We would like to thank the reviewers for engaging with our responses, even under these unusual circumstances.

Across the reviews key overlapping concerns emerged:

Reviewers asked for clearer justification of why certain benchmarks align with System1 or System2 reasoning. We clarify that our mapping was grounded in established cognitive science literature, but we recognize that benchmarks may draw on both reasoning styles to varying degrees. The classification is intended as a guiding tendency rather than a strict, one-to-one mapping. We will emphasize the nuanced about the mapping in our revision.

Reviewers are concerned about the moderate size of the dataset and some examples appear stylized. As mentioned in our rebuttal, the stylized reasoning paths were a deliberate design choice. Recent NLP studies have demonstrated that small, carefully curated, and expert-verified datasets can be highly effective for aligning LLM behavior.

Reviewers noted that the results might be intuitive. While some findings may align with intuition, our work goes beyond validation by systematically training and evaluating LLMs under explicit cognitive-style constraints, revealing predictable, tunable trade-offs in accuracy, efficiency, and uncertainty. To our knowledge, this is the first reproducible framework linking dual-process theory to LLM alignment.

Reviewers suggested dynamic switching between reasoning styles. While this paper focuses on establishing a foundation by aligning LLMs independently to System1 and System2 reasoning styles and analyzing their distinct strengths and trade-offs, we agree that dynamic reasoning represents a valuable next step but out of the scope of this paper (lines 637-639).

Regarding the reviewer concerns about the risks of reinforcing harmful biases when aligning LLMs with System1 reasoning. Our use of heuristics is aimed at improving efficiency rather than justifying or amplifying bias.

---

### Decision · Program_Chairs · 2025-09-17

**Decision:**

Reject

**Comment:**

Note that the rebuttals for this paper were not posted in accordance with the directions and at the direction of the NeurIPS program chairs they were ignored. While the authors did post several comments on this asking for a change of rules this is not allowed in fairness to all submissions.

This led to no or very little interaction with the rebuttals and this Meta Review is only considering the main reviews as submitted.

As such, the reviews are clear in their direction that this paper is not ready for publication at NeurIPS. While all reviewers liked the motivation, and taking inspiration from the S1/S2 models is a popular course of research, there are key shortcomings in the current paper, specifically:

- Connection to benchmarks needs to be improved as well as overall evaluations to more clearly demonstrate improvements.

- Resulting interpret-ability claims need to be improved in light of better bench-marking.

- Many claims are "heuristically recognized" but not clearly demonstrated, these need to be more clearly grounded.

Note that all reviews contain questions and guidance for updating and refining the paper and we strongly encourage the authors to take these into account with revising the paper.